# ATTACKING DEEP NETWORKS WITH SURROGATE-BASED ADVERSARIAL BLACK-BOX METHODS IS EASY

**Nicholas A. Lord, Romain Mueller & Luca Bertinetto**
www.five.ai
{nick,romain.mueller,luca.bertinetto}@five.ai

## ABSTRACT

A recent line of work on black-box adversarial attacks has revived the use of transfer from surrogate models by integrating it into query-based search. However, we find that existing approaches of this type underperform their potential, and can be overly complicated besides. Here, we provide a short and simple algorithm which achieves state-of-the-art results through a search which uses the surrogate network's class-score gradients, with no need for other priors or heuristics. The guiding assumption of the algorithm is that the studied networks are in a fundamental sense learning similar functions, and that a transfer attack from one to the other should thus be fairly "easy". This assumption is validated by the extremely low query counts and failure rates achieved: e.g. an untargeted attack on a VGG-16 ImageNet network using a ResNet-152 as the surrogate yields a median query count of 6 at a success rate of 99.9%. Code is available at https://github.com/fiveai/GFCS.

## 1 INTRODUCTION

The paper that introduced adversarial examples in computer vision (Szegedy et al., 2014) also initiated the study of their transfer across models. This directly yielded two parallel lines of research into "white-box" and "black-box" attacks on classification systems. The white-box attacks (e.g. Goodfellow et al. (2015); Moosavi-Dezfooli et al. (2016); Carlini & Wagner (2017)) assumed full knowledge of the victim model architecture and parameters, and would typically exploit the analytical gradients of the network outputs with respect to the input image. These methods were primarily concerned with demonstrating the *existence* of adversarial examples, as well as optimising criteria such as their norms or the time spent computing them. The earliest black-box attacks (e.g. Papernot et al. (2017); Liu et al. (2017)), on the other hand, assumed no access to the attacked model beyond an end user's ability to input images and receive output classifications. They sought to produce adversarial examples on the victim model by transferring them from a known surrogate model. This surrogate may have represented a model trained separately on a comparable problem or one trained to mimic the victim through a sequence of online queries; the transfer attack would typically comprise something as simple as a single step in the surrogate gradient direction. These methods were concerned primarily with the *discoverability* and *predictability* of adversarial examples. The key assumption underlying this branch of study was that different ML architectures, when trained on sufficiently similar problems, would exhibit similar adversarial vulnerabilities at the same inputs. As has become better understood and explained since (Olah et al., 2017; Jetley et al., 2018; Ilyas et al., 2019b), this is equivalent to those networks learning similar feature responses to one another, i.e., learning similar solutions to the problem. Such attacks did demonstrate nontrivial success rates, and thus, the partial validity of that hypothesis. However, the field widely acknowledged that there were limits to their reliability, and sought alternatives with higher success rates.

To this end, Chen et al. (2017) and Bhagoji et al. (2018) introduced a modification of the threat model: they assumed the victim to provide not only the top class prediction, but also the candidate class scores. This relaxation eliminated the need for a surrogate, enabling the use of query-based methods to numerically estimate the victim's gradients directly. These approaches and their many descendants are called "score-based" attacks. The common problem they all face, noted in the seminal works above, is that direct numerical estimation of gradients is linear in the dimension of the input space. Depending on the input image resolution, this ranges from being costly to being infeasible. Thus, a core issue in score-based attacks is the need to limit the query count without compromising the

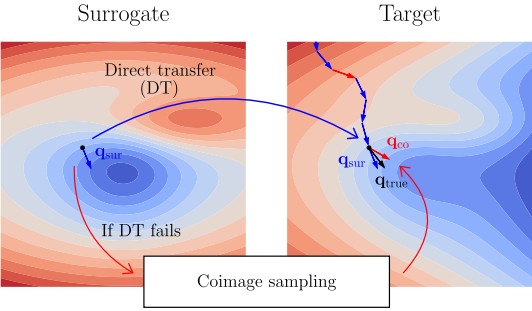

Figure 1: Transfer of information from a surrogate to a target in GFCS: Gradient First, Coimage Second. A sequence of optimisation steps is taken through the loss landscape of the target model, as shown on the right. Each candidate direction $\mathbf{q}$ is supplied by the surrogate in one of two ways: either through direct transfer of the surrogate's own gradient $\mathbf{q}_{sur}$, or as a $\mathbf{q}_{co}$ generated through a process of coimage sampling described in Sec. 2.2. $\mathbf{q}_{true}$ is the target's true but inaccessible gradient. As shown in Sec. 3.2, the method relies mostly on the directly transferred $\mathbf{q}_{sur}$s, using the $\mathbf{q}_{co}$s to avoid failure.

quality of the gradient estimate: in Chen et al. (2017) alone, proposals included coordinate descent, basis transformation, coarse-to-fine optimisation, and priors on the gradient. The line of work which followed (Ilyas et al., 2018; 2019a; Li et al., 2019; Liu et al., 2020; Guo et al., 2019b; Tu et al., 2019) explored ways to improve the fundamental approach by experimenting with different optimisation methods and priors. Strikingly, and perhaps surprisingly, one of the most successful efforts was SimBA (Guo et al., 2019a), for "Simple Black-Box Attack". Its name refers to the fact that it is an especially basic form of coordinate descent which achieves competitive results despite its deliberately simple approach: the greedy sequential assignment of signs to fixed-length steps along orthogonal basis directions.

Though query-driven score-based black-box attacks were proposed as an alternative to transfer methods, the two approaches are not incompatible with one another. They are actually complementary: a query-based strategy can benefit from search directions that are more promising *a priori*, and a transfer-based strategy can benefit from a flexible optimiser that allows it to dynamically correct approximation errors and consider alternative hypotheses. This was first suggested in Cheng et al. (2019) and Guo et al. (2019b), which were followed by Tashiro et al. (2020) and Yang et al. (2020). In this paper, we argue that the core intuition of this branch of literature is sound. However, we also argue that when combining transfer- and query-based approaches, it is crucial to recognise that the failure of surrogate gradient transfer is generally overestimated, even by approaches which leverage it. Transfer typically succeeds: it just requires an occasional source of appropriately chosen alternative hypotheses within a sensible optimisation framework. It is this core insight that enables us to propose a simple but powerful new algorithm which achieves state-of-the-art results against the most modern competing methods on this problem, under their own experimental setup of minimising the number of queries required to identify $\ell_2$-norm-bounded adversarial perturbations on specific networks. We call our approach "GFCS: Gradient First, Coimage Second", as its search directions represent either the surrogate gradient of the adversarial loss itself, or, failing that, a random choice from the row space of the surrogate Jacobian (called the "coimage"). The former case represents a standard transfer attack; the latter involves searching the space of features to which the locally linearised surrogate exhibits any response at all, and is itself a generalised form of gradient transfer. The optimisation method is just a SimBA (Guo et al., 2019a) variation within a standard projected gradient ascent (PGA) context. Key to the method's efficiency is the identification of effective local search spaces of low dimension: in the case of the common ImageNet Inception-v3 implementation, confining the search to the coimage reduces the dimension from $299 \cdot 299 \cdot 3$ (the input resolution) to $1000$ (the number of output classes). The loss gradient is of course one-dimensional.

Whether a threat model which assumes access to both class scores and surrogates is realistic from a security perspective is a controversial question, and one we are agnostic to. Our interest in the problem is analytical, and we have two main points to make. First, if this threat model is to be considered to be of interest (as it has been in the prior art), then it should be understood just how "easy" the problem as currently posed is: even a single surrogate reduces the query count to a handful. Second, the fact that GFCS performs as it does while relying entirely on gradient transfer between networks demonstrates that those networks are, in an important sense, very similar to one another.

## 2 METHOD

### 2.1 PRELIMINARIES

The adversarial attack problem appears in slightly different variations. Generally, given a classifier function $\mathbf{f}(\mathbf{x})\colon \mathcal{X} \to \mathcal{Y}$, the goal is to supply an adversarial example $\mathbf{x}_{\text{adv}} \in \mathcal{X}$ which, while "near" to a given $\mathbf{x}_{\text{in}}$ under some definition, is such that $\mathbf{f}(\mathbf{x}_{\text{adv}})$ differs from $\mathbf{f}(\mathbf{x}_{\text{in}})$ in a particular noteworthy way. In image classification, we typically have the situation that $\mathcal{X} \subseteq \mathcal{R}^D$ and $\mathcal{Y} \subseteq \mathcal{R}^C$, with $\mathbf{f}$ a network mapping a $D$-dimensional input image to a $C$-dimensional class-score/logit vector. The *untargeted* attack objective is $\operatorname{argmax}_c \mathbf{f}_c(\mathbf{x}_{\text{adv}}) \neq \operatorname{argmax}_c \mathbf{f}_c(\mathbf{x}_{\text{in}})$, i.e. that the top predicted class of $\mathbf{x}_{\text{adv}}$ is different from that of $\mathbf{x}_{\text{in}}$. The *targeted* attack objective is instead $\operatorname{argmax}_c \mathbf{f}_c(\mathbf{x}_{\text{adv}}) = t$, meaning that the net predicts a specific target class $t$ other than the original prediction $\operatorname{argmax}_c \mathbf{f}_c(\mathbf{x}_{\text{in}})$.

As the subfield has expanded to consider perturbation classes including non-rigid deformation (Xiao et al., 2018), semantically insignificant distortion (Hosseini & Poovendran, 2018; Brown et al., 2018), and movement within the estimated natural image manifold (Zhao et al., 2018; Hendrycks et al., 2021), corresponding definitions of adversarial perturbation magnitude have been adopted, including total variation and manual human assessment. The most common measure, used by all methods to be compared against in this paper, is $\|\mathbf{x}_{\text{in}} - \mathbf{x}_{\text{adv}}\|_p$, with $p \in \{2, \infty\}$: here, we use $p = 2$.

In some experimental setups, the attacker's goal is to satisfy the adversarial objective while minimising the perturbation magnitude. In others, it is to minimise the number of network evaluations performed while finding any adversarial example with a perturbation magnitude within a pre-specified bound. In black-box attacks, the latter setup is common, with "evaluations" being defined as queries to the victim model: this is likewise our focus.

### 2.2 ALGORITHM

---

**Algorithm 1** GFCS: Gradient First, Coimage Second

---

1: **Input:** A targeted image $\mathbf{x}_{\text{in}}$, loss function $L$, a victim classifier $\mathbf{v}$, a set of surrogate models $\mathcal{S}$, a step length $\epsilon$, and a norm bound $\nu$.
2: **Output:** adversarial image $\mathbf{x}_{\text{adv}}$ within distance $\nu$ of $\mathbf{x}_{\text{in}}$
3: $\mathbf{x}_{\text{adv}} \leftarrow \mathbf{x}_{\text{in}}$
4: $\mathcal{S}_{\text{rem}} \leftarrow \mathcal{S}$
5: **while** $\mathbf{x}_{\text{adv}}$ is not adversary **do**
6:     **if** $\mathcal{S}_{\text{rem}} \neq \emptyset$ **then**
7:         Randomly sample surrogate model $\mathbf{s}$ from $\mathcal{S}_{\text{rem}}$
8:         $\mathcal{S}_{\text{rem}} \leftarrow \mathcal{S}_{\text{rem}} \setminus \mathbf{s}$
9:         $\mathbf{q} \leftarrow \frac{\nabla_{\mathbf{x}} L_{\mathbf{s}}(\mathbf{x}_{\text{adv}})}{\|\nabla_{\mathbf{x}} L_{\mathbf{s}}(\mathbf{x}_{\text{adv}})\|_2}$
10:    **else**                  ▷ None of the surrogate loss gradients work, so revert to ODS.
11:         Randomly sample surrogate model $\mathbf{s}$ from $\mathcal{S}$
12:         Sample $\mathbf{w} \sim U(-1, 1)^C$
13:         $\mathbf{q} \leftarrow \mathbf{d}_{\text{ODS}}(\mathbf{x}_{\text{adv}}, \mathbf{s}, \mathbf{w})$               ▷ See Eqn. 1 for definition.
14:     **for** $\alpha \in \{\epsilon, -\epsilon\}$ **do**
15:         **if** $L_{\mathbf{v}}(\Pi_{\mathbf{x}_{\text{in}}, \nu}(\mathbf{x}_{\text{adv}} + \alpha \cdot \mathbf{q})) > L_{\mathbf{v}}(\mathbf{x}_{\text{adv}})$ **then**
16:            $\mathbf{x}_{\text{adv}} \leftarrow \Pi_{\mathbf{x}_{\text{in}}, \nu}(\mathbf{x}_{\text{adv}} + \alpha \cdot \mathbf{q})$
17:            $\mathcal{S}_{\text{rem}} \leftarrow \mathcal{S}$   ▷ Reset candidate surrogate set to input set; resume using loss gradients.
18:            **break**

---

The entirety of the proposed method is given in pseudocode as Algorithm 1. As indicated in Sec. 2.1, the method takes a victim classifier $\mathbf{v}$, an input image $\mathbf{x}_{\text{in}}$, and a norm bound $\nu$. The projection operator $\Pi_{\mathbf{x}_{\text{in}}, \nu}$ confines its input to the $\nu$-ball centred on $\mathbf{x}_{\text{in}}$: its inclusion in the algorithm represents a standard projected gradient ascent (PGA) implementation. Additionally, the method requires a loss function $L$, a set $\mathcal{S}$ of one or more surrogate models, and a step length $\epsilon$ representing the fixed length of the perturbations to be attempted at each iterate[1] along its candidate direction. The loss $L$ can be any function of the iterate that serves as a suitable proxy for the adversarial objective, as in Sec. 2.1:

---

[1]In optimisation, this term refers to any given intermediate value of the variable being optimised.

the only requirement here is that it be once differentiable. In our implementation, we make the popular and effective choice of the margin loss $L_\mathbf{f}(\mathbf{x}) = \mathbf{f}_{c_t}(\mathbf{x}) - \mathbf{f}_{c_s}(\mathbf{x})$ where $c_s = \mathrm{argmax}_c \, \mathbf{v}_c(\mathbf{x})$ and $c_t = \mathrm{argmax}_{c \neq c_s} \mathbf{v}_c(\mathbf{x})$, i.e. the difference between the highest and second-highest (or, in the targeted case, the target) class scores. Note that the class IDs $c_t$ and $c_s$ are defined by the ranking according to $\mathbf{v}$, but are evaluated on the net $\mathbf{f}$ parametrising the loss, which is either $\mathbf{s}$ or $\mathbf{v}$ depending on the line of the algorithm (9 and 15, respectively). The natural assumption of a surrogate method is that the surrogate will provide useful information about the victim, but there is no "hard" requirement on the surrogates $\mathbf{s} \in \mathcal{S}$ other than being once-differentiable functions mapping $\mathcal{X} \to \mathcal{Y}$. The definition of ODS direction $\mathbf{d}_{\mathrm{ODS}}$ for network $\mathbf{f}$ and input $\mathbf{x}$ is, as in Tashiro et al. (2020),

$$\mathbf{d}_{\mathrm{ODS}}(\mathbf{x}, \mathbf{f}, \mathbf{w}) = \frac{\nabla_\mathbf{x}(\mathbf{w}^\intercal \mathbf{f}(\mathbf{x}))}{\|\nabla_\mathbf{x}(\mathbf{w}^\intercal \mathbf{f}(\mathbf{x}))\|_2} = \frac{\mathbf{w}^\intercal(\nabla_\mathbf{x}\mathbf{f}(\mathbf{x}))}{\|\nabla_\mathbf{x}(\mathbf{w}^\intercal \mathbf{f}(\mathbf{x}))\|_2}, \tag{1}$$

where $\mathbf{w}$ is sampled from the uniform distribution over $[-1, 1]^C$. By definition, it is the normalised gradient of a randomly weighted sum of all of the class scores. Equivalently, by linearity, it is a randomly weighted sum of all of the class-score gradients (i.e. rows of the Jacobian matrix), which are themselves a basis of the coimage of the linear approximation of $\mathbf{f}$: the subspace which $\mathbf{f}$ exhibits any nonzero response to.

As indicated in Sec. 1, the logic of the method is simple. At any given iterate, the method tries to proceed in a SimBA-like manner by testing the change in adversarial loss at fixed-length steps along candidate directions, projected back into the feasible set where necessary. It does so exclusively using normalised loss gradients from the surrogates in the input set (drawn in random order, without replacement), unless and until it has exhausted them at that iterate without success. As we will demonstrate in Sec. 3.2, this intermediate failure state is seldom reached. If this state is reached, however, the method instead randomly samples a surrogate (with replacement) and an ODS direction from that surrogate, attempting a SimBA update each time, until an improvement in the loss is realised. Once such a successful update occurs, the method resets the candidate surrogate set to the input set and resumes using normalised loss gradients only. The method terminates on finding an adversarial example or on exceeding an upper bound on the query count if one has been specified.

## 3 EXPERIMENTS

### 3.1 UNTARGETED ATTACKS

**Experimental setup:** We compare GFCS to the methods of Cheng et al. (2019); Tashiro et al. (2020); Yang et al. (2020) by designing an experimental framework covering the key aspects of the original experiments in the respective source works. We use each method to perform $\ell_2$-norm-constrained untargeted attacks against the same 2000 randomly chosen correctly classified ILSVRC2012 validation images per victim network. A maximum query count of 10000 is set per example (beyond which failure is declared), and the $\ell_2$ bound (enforced using PGA) is set to the commonly chosen $\sqrt{0.001D}$, where $D$ is the image dimension in the victim network's native input resolution. The victim networks are the commonly chosen VGG-16, ResNet-50, and Inception-v3. The experiments are repeated for each of two choices of surrogate set: ResNet-152 alone (as in Cheng et al. (2019); Yang et al. (2020)), and the set {VGG-19, ResNet-34, DenseNet-121, MobileNet-v2}, as in Tashiro et al. (2020). The latter is omitted for LeBA (Yang et al., 2020), which does not admit non-singleton surrogate sets. All networks used are pretrained models available via PyTorch/torchvision. Parameter values of competitors are as they specify except where we note otherwise for reasons that will be discussed below. LeBA is run in "train" mode on a held-out set of 1000 images and then evaluated in "test" mode on the same set of 2000 used for all other methods. P-RGF always uses the adaptive coefficient mode. P-RGF and ODS-RGF are based on our own PyTorch port of the reference P-RGF code, which will be released along with this paper: no public implementation of ODS-RGF currently exists otherwise. We include the surrogate-free (Andriushchenko et al., 2020) for comparison.

**Results:** Table 1 reports attack success rates and median query counts, and Fig. 2 plots cumulative success counts against the maximum queries spent per example (CDFs, modulo normalisation). Unlike prior work, we do not report means, as these are inappropriate for summarising the long-tailed distributions resulting from these methods. Uncertainty is represented as standard error in Table 1 and by 95% confidence intervals in Fig. 2, in both cases having been obtained via bootstrap sampling. Two things are readily apparent in Table 1. First, *all* of the studied methods have very high success rates on

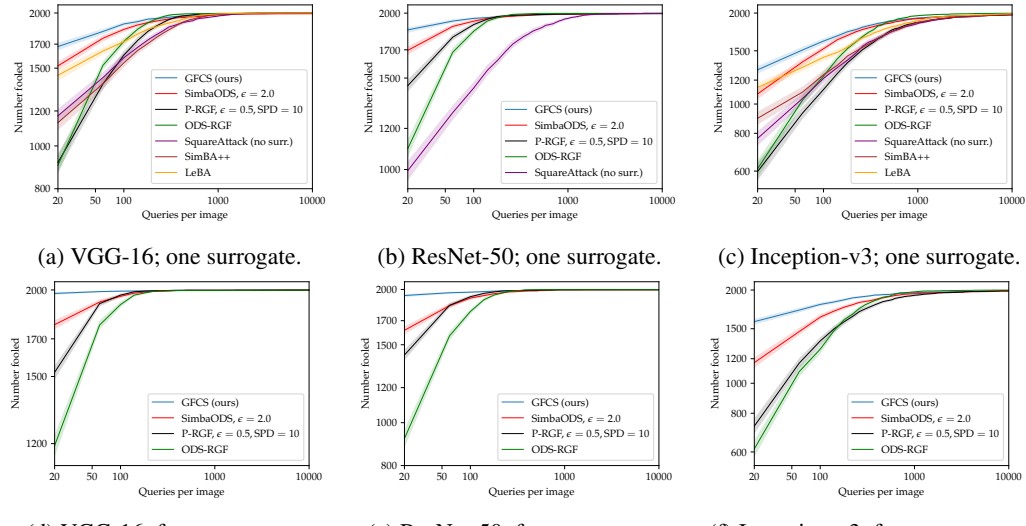

(a) VGG-16; one surrogate.  (b) ResNet-50; one surrogate.  (c) Inception-v3; one surrogate.

(d) VGG-16; four surrogates.  (e) ResNet-50; four surrogates.  (f) Inception-v3; four surrogates.

Figure 2: CDFs representing the number of successfully attacked examples at different query counts when performing untargeted black-box attacks on VGG-16, ResNet-50, and Inception-v3 networks.

Table 1: Median query count for state-of-the-art untargeted black-box methods that make use of surrogates. The missing entry at † indicates the LeBA code crashing before completing the full set of images for ResNet-50.

| Surrogates | Method | Median queries [Success rate] | | | | | |
|---|---|---|---|---|---|---|---|
| | | VGG-16 | | ResNet-50 | | Inception-v3 | |
| | SimBA-ODS | $117 \pm 5.1$ | [99.45%] | $91.5 \pm 3.2$ | [99.65%] | $275 \pm 14$ | [95.10%] |
| | SimBA-ODS $\epsilon=2$ | $15 \pm 0.7$ | [99.65%] | $11 \pm 0.4$ | [99.90%] | $36 \pm 1.9$ | [98.50%] |
| | P-RGF | $128 \pm 1.1$ | [99.95%] | $62 \pm 1.2$ | [100%] | $282 \pm 12$ | [99.25%] |
| 1 | P-RGF $\epsilon=0.5$, SPD=10 | $48 \pm 2.0$ | [99.90%] | $16 \pm 0.0$ | [99.95%] | $90 \pm 4.9$ | [99.00%] |
| | ODS-RGF | $44 \pm 0.0$ | [99.95%] | $33 \pm 0.0$ | [100%] | $77 \pm 2.4$ | [99.95%] |
| | SimBA++ | $30 \pm 3.0$ | [100%] | $5 \pm 0.5$ | [100%] | $59 \pm 3.4$ | [99.4%] |
| | LeBA | $8 \pm 0.74$ | [100%] | —† | | $27 \pm 3.1$ | [99.45%] |
| | **GFCS (ours)** | $\mathbf{6 \pm 0.3}$ | [99.90%] | $\mathbf{4 \pm 0.4}$ | [99.85%] | $\mathbf{18 \pm 1.1}$ | [98.60%] |
| | GF, no CS (ablation) | —"— | [58.55%] | —"— | [75.90%] | (failure) | [38.25%] |
| | SimBA-ODS | $68 \pm 2.2$ | [99.90%] | $108.5 \pm 4.5$ | [99.80%] | $227 \pm 9.9$ | [96.65%] |
| | SimBA-ODS $\epsilon=2$ | $10 \pm 0.3$ | [100%] | $14 \pm 0.5$ | [99.90%] | $29 \pm 1.4$ | [100%] |
| | P-RGF | $64 \pm 0.6$ | [99.95%] | $66 \pm 0.5$ | [100%] | $232 \pm 4.7$ | [99.15%] |
| 4 | P-RGF $\epsilon=0.5$, SPD=10 | $16 \pm 2.3$ | [100%] | $24 \pm 1.2$ | [100%] | $60 \pm 2.1$ | [99.60%] |
| | ODS-RGF | $33 \pm 0.0$ | [100%] | $44 \pm 0.0$ | [100%] | $77 \pm 5.5$ | [100%] |
| | **GFCS (ours)** | $\mathbf{4 \pm 0.2}$ | [100%] | $\mathbf{4 \pm 0.0}$ | [99.95%] | $\mathbf{9 \pm 0.5}$ | [99.40%] |
| | GF, no CS (ablation) | —"— | [98.65%] | —"— | [96.50%] | —"— | [80.20%] |

this problem, against all of the victim networks: the lowest rate observed is 95.10% for SimBA-ODS on Inception-v3 with ResNet-152 as the lone surrogate. Second, GFCS incurs an *extremely* low median query count while achieving a similarly high success rate to all other methods. This fact can be seen in more detail in the single-surrogate results of Figs. 2a, 2b, and 2c, in which GFCS clearly dominates the low-query regime, and even more strikingly so in the multi-surrogate Figs. 2d, 2e, and 2f. This is despite GFCS's simplicity: compare Alg. 1 against, for example, LeBA's training of its surrogate in a separate step. Note that our choice of SimBA-ODS as the coimage sampler is partly about simplicity: as the results demonstrate, a very small number of failures are expected when it is used on its own, and we effectively inherit them. At the cost of a bit of added complexity in the implementation, e.g. ODS-RGF could be substituted, and would likely lead to further improvements in the failure rates, while still representing a form of GFCS. There is an additional phenomenon of note: Table 1 also demonstrates the dependence of the performance of existing methods on their own choices of parameter values. Strikingly, most of the empirical benefit of ODS-RGF over the earlier P-RGF is due to the different choice of default parameters in the respective methods: when P-RGF simply uses the default parameters of ODS-RGF, it actually considerably outperforms it in terms of median query count on ResNet-50, at the cost of a 0.05% increase in the single-surrogate failure rate.

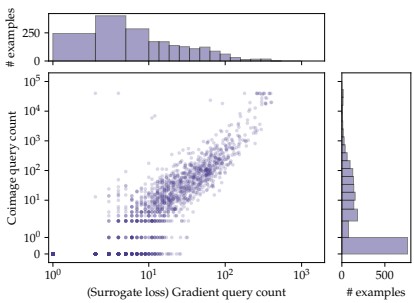 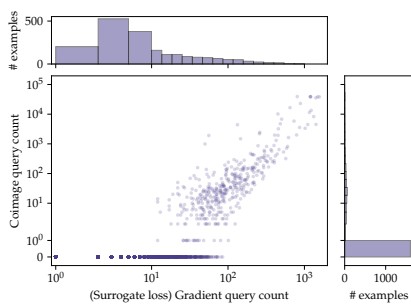

(a) Inception-v3; one surrogate.  (b) Inception-v3; four surrogates.

Figure 3: Breakdown of the total query count for the proposed method, *GFCS: Gradient First, Coimage Second*. The x-axis represents the number of queries for a successful attack required by the *gradient* part of the method, and the y-axis the number required by the *coimage* part. Histograms on the top and right sides of the scatter plots represent marginal empirical distributions.

SimBA-ODS benefits dramatically from an order-of-magnitude increase in its step-size parameter. This, in and of itself, serves as further evidence of our central thesis that transfer is generally pursued too timidly in this context. Of course, one can be *too* aggressive: see the ablation lines in the table representing the exclusive use of loss gradients, i.e. "GF without the CS".

## 3.2 ANALYSIS OF ALGORITHM BEHAVIOUR

To delve deeper into the results of Sec. 3.1, we plot each attacked example as a 2D point whose $x$-coordinate is the number of queries expended by the surrogate loss gradient block of the algorithm, and whose $y$-coordinate is the analogous count for the coimage block. This gives the scatter plot of Fig. 3, which is supplemented by marginal histograms corresponding to the axes opposite them. The figure shows results obtained using Inception-v3 as the victim: see Appendix A.3 for analogous figures for VGG-16 and ResNet-50. Note that the axes of the main scatter plots are log-log, while those of the marginal histograms are linear-log. Some phenomena are readily evident. For one, there is a large fraction of examples (represented by the dense horizontal rows of dots at the bottom of the plots) that succeed within a very low number of queries (on the order of 1-10), which are entirely or almost entirely due to the surrogate gradient transfer, with ODS used seldom or not at all. As these low-query clusters are extremely dense, the corresponding marginals should be consulted (best under zoom) in order to quantify them. For another, the number of examples outside of this regime falls considerably when the four-surrogate set is used instead of ResNet-152 on its own, as can be seen by comparing the left and right sides of the figure. It is clear that the examples that rely on the interplay between the gradient- and coimage-based direction generators are reduced to a nontrivial (i.e. sufficient to affect the failure rate if not handled) but nonetheless relatively small group. Overall, there is an order-of-magnitude difference between surrogate loss gradient queries and ODS queries in the points extending away from the dense low-query cluster at the bottom, i.e. the examples that rely on both submethods. That is, when the ODS block is required, it typically requires far more queries to progress the optimiser than in the much more common cases in which the gradient suffices.

## 3.3 ON THE IMPORTANCE OF INPUT-SPECIFIC PRIORS

We have demonstrated that surrogate CNNs are *sufficient* to craft extremely effective score-based black-box attacks on other CNNs trained on the same problem. We now make an empirical argument that such surrogates are likely *necessary* in achieving this level of performance. That is, the location-specific information encoded by a surrogate network, in which the gradient prior is a function of the input image-space point, increases attack effectiveness beyond that available to methods that use location-independent priors and infer location-specific properties online. We do not consider this to be a trivial point. The phenomenon of universal adversarial perturbations (Moosavi-Dezfooli et al., 2017) demonstrates a level of location agnosticism in adversarial vulnerability, and the original SimBA method (Guo et al., 2019a) on which GFCS is based is itself a demonstration of the remarkable effectiveness of online determination of the signs of predetermined basis vectors. The latter paper

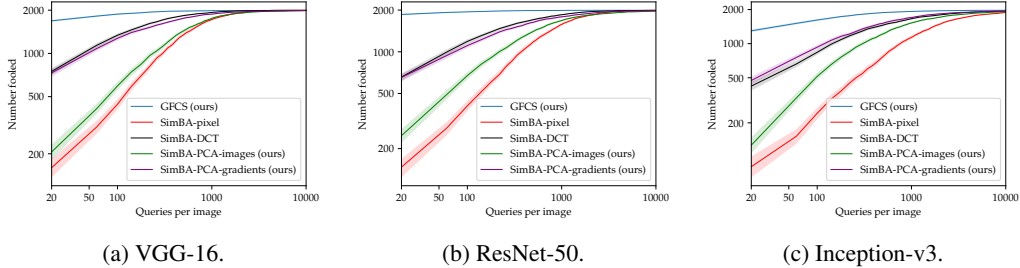

(a) VGG-16.  (b) ResNet-50.  (c) Inception-v3.

Figure 4: Comparison between GFCS, which evaluates surrogate gradients locally, vs. flavours of SimBA which build location-agnostic gradient priors. Please refer to Sec. 3.3.

itself proposed, as future work, "to further investigate the selection of different sets of orthonormal bases, which could be crucial to the efficiency of [the] method by increasing the probability of finding a direction of large change." *A priori*, it is unclear what the added benefit of a surrogate network will be, vs. a well-chosen adversarial subspace with a prior ordering on its basis vectors.

To study this within our SimBA-based optimisation context, we propose a variant called "SimBA-PCA", which is defined by the approach it takes to generating the SimBA basis matrix. Following the method used in Moosavi-Dezfooli et al. (2017); Jetley et al. (2018), we gather adversarial examples[2] over a sample input set as columns in a matrix, then compute that matrix's SVD. That is, we produce the ordered set of vectors that would be expected to represent the $\ell_2$-optimal basis for reproducing the adversarial examples on the given set, when constrained to follow SimBA's iterative adversary-building optimisation procedure. We then compare this against SimBA using both the canonical pixel and DCT bases in random order, and GFCS. An an interesting aside, we also include the result of using the procedure on raw images directly, i.e. using the principal components of the input data as the search directions. All of these methods are placed within the same norm-bounded PGA framework.

Fig. 4 demonstrates the results of this experiment. As predicted, the gradient-based basis ("SimBA-PCA-gradients") comfortably outperforms the canonical pixel basis ("SimBA pixel"), as well as the principal image components ("SimBA-PCA-images"), the latter fact indicating that the information encoded in adversarial directions indeed goes beyond simple correspondence to modes of data variation. Note, though, that the principal data directions do outperform the naïve pixel basis, unsurprisingly revealing nontrivial correlation between modes of data variation and features learned by the network. However, despite forming a "natural adversarial basis", the adversarial singular vector matrix does *not* generally outperform the DCT basis: their results are very close, with the DCT basis sometimes slightly outperforming. That is, the DCT basis with a suitably tuned frequency count appears to already represent the limit of what a SimBA-style iteration through a single fixed orthonormal basis can accomplish, and the SimBA-PCA procedure has at best recovered an equivalent to it. GFCS, on the other hand, is far more efficient than all of the compared methods, demonstrating the performance that is available when it is possible to condition the prior on the iterate, i.e. to use local gradient information. This is one way of viewing why it is that the use of surrogates is as powerful as it is. This is despite the fact that these surrogates, rather than having been originally designed to simulate their victims, were actually supposed to have represented architectural alternatives to them.

## 3.4 TARGETED ATTACKS

We also test our method in the targeted attack scenario. The setup is the same as in Sec. 3.1, except as noted otherwise. For these experiments, instead of the margin loss, we use the log loss of the target-class softmax score, as is often chosen for targeted attacks: $L_{\mathbf{f}}(\mathbf{x}) = \log p_t$, where $p_t = \frac{e^{\mathbf{f}_t(\mathbf{x})}}{\sum_c e^{\mathbf{f}_c(\mathbf{x})}}$. That is, the goal of the attacker is to push the target-class confidence up at the expense of all other classes generally, rather than suppressing a specific source class. We perform multiple runs for each setting, with the target class for each image chosen at random (uniformly) from all ImageNet classes other than the ground truth. Neither Cheng et al. (2019) nor Yang et al. (2020) provide results or code for targeted attacks. While Tashiro et al. (2020) report targeted attack numbers for both P-RGF and

---

[2]We have experimented with various methods of generating the source adversaries, all of which yield indistinguishable results. Here, we simply use the gradients of random class scores, a basic targeted FGM.

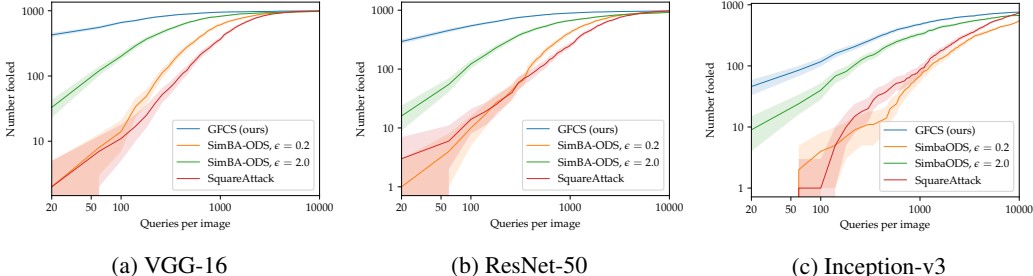

(a) VGG-16          (b) ResNet-50          (c) Inception-v3

Figure 5: CDFs showing the number of successfully attacked examples at different query counts when performing targeted black-box attacks on 1000 ImageNet images, with four surrogates.

their own ODS-RGF, no supporting implementation is available. As noted before, our own P-RGF and ODS-RGF results in Sec. 3.4 were produced using a port of the public P-RGF repository, which only supports untargeted attacks (a restriction that carries through to the ODS-RGF modification we have made to it). We thus work with the provided SimBA-ODS implementation, comparing it in isolation to GFCS, in which it serves as the backup method. The results are displayed in Fig. 5. As can be seen, even under this more difficult attack scenario, under which the more aggressive transfer strategy of GFCS might be expected to suffer, our method retains its considerable advantage.

While the targeted attack method of Huang & Zhang (2020) is both powerful and conceptually related, it is architecturally tied to the $\ell_\infty$ norm in its given formulation, and thus not appropriate for our $\ell_2$-bound comparisons. Further, given a trained surrogate, the method requires that one *then* train the method's adversarial encoder-decoder network on a held-out validation set: in the case of targeted attacks, this must be done *per target class*, as acknowledged by the authors in the original work. Our approach, by contrast, uses the surrogate directly without any issue, on any target class.

## 4  RELATED WORK

Cheng et al. (2019) and Guo et al. (2019b) were the first to, in their words, "bridge the gap between the transfer-based attacks and the query-based ones" in the score-based black-box context, noting that earlier methods "often suffer from low attack success rates (in the transfer case) or poor query efficiency (in the query-based case) since it is non-trivial to estimate the gradient in a high-dimensional space with limited information". Guo et al. (2019b) did this by supplying a bandits optimiser (as used in Ilyas et al. (2019a)) with surrogate gradient estimates at each iteration, diversifying the proposed search directions through the use of test-time dropout and drop-layer. Cheng et al. (2019) presented a variant of random gradient-free (RGF) optimisation using the surrogate gradient as a biasing prior (P-RGF) and featuring a dynamically estimated bias parameter, albeit one whose optimal value itself depends on the unknown target gradient. The Output-Diversified Sampling (ODS) approach of Tashiro et al. (2020) sampled gradients of randomly weighted logit sums over multiple surrogate models in order to generate search directions for variations on SimBA, RGF, and some Boundary Attack (Brendel et al., 2018) variants, demonstrating improvement over all of the base methods. Yang et al. (2020) also proposed a surrogate-enhanced version of SimBA ("SimBA++"), sampling candidate pixels to perturb according to the distribution specified by the corresponding magnitudes of the surrogate gradient components, as well as making periodic attempts to directly transfer surrogate gradients estimated using smoothing and momentum. When combined with "High Order Gradient Approximation", a method for dynamically updating the surrogate by matching the first- and zeroth-order victim model behaviour, it is dubbed "Learnable Black-Box Attack" (LeBA).

In addition to the above, we review relevant works on black-box attacks, particularly score-based methods that use (possibly learned) prior information and/or alternative optimisers. When they feature a relevant concept, we also include variations of the Boundary Attack method of Brendel et al. (2018), which avoid requiring scores in exchange for much higher query counts:

**Frequency/scale priors:** As noted by (Chen et al., 2017; Bhagoji et al., 2018), in order to be practical, any black-box method that relies on gradient estimation must have an approach to effectively reduce the intrinsic dimension of the estimate space. Guo et al. (2019a; 2020) use low-frequency DCT dimensions for this purpose. The Boundary Attack variant of Brunner et al. (2019) uses Perlin noise similarly, and the Gaussian smoothing of the gradient in Yang et al. (2020) is conceptually related. Also closely related to the low-frequency prior is the use of spatial downsampling, whether applied

to the image or its gradient, and whether implemented through pooling, interpolation, or striding: Tu et al. (2019); Ilyas et al. (2019a); Li et al. (2019); Wang et al. (2021) all involve a version of it. Coarse-to-fine approaches, in which results at a coarser scale either solve the problem or initialise the optimiser of a finer one, appear in Moon et al. (2019); Al-Dujaili & O'Reilly (2020).

**Data distribution modelling:** Li et al. (2019) assumes the ability to parametrically model an adversarial distribution in the vicinity of the input point. Dolatabadi et al. (2020) replaces this parametric model with a normalising flow model trained on clean data. Tu et al. (2019) represents the data using an autoencoder and conducts the attack in its latent space. Huang & Zhang (2020) likewise conducts a latent-space attack, but in that of an encoder-decoder network trained to output adversaries on a source network. Ru et al. (2019) attempts to learn the latent-space dimension itself.

**Attention:** Wang et al. (2021) uses the output of CAM (Zhou et al., 2016) on a proxy net as a map of pixels to attack. In Brunner et al. (2019), a similar map is derived from the difference between the adversarial and original images, and used to weight the sampled perturbation elementwise.

**Gradient/feature priors:** The use of gradient priors in Guo et al. (2019b); Cheng et al. (2019); Tashiro et al. (2020); Yang et al. (2020) is discussed above, and the transfer-based approaches Papernot et al. (2017); Liu et al. (2017) are of course fundamentally based on this. Besides these, the method of Brunner et al. (2019) uses the gradient of a surrogate model to bias the orthogonal perturbation used in Boundary Attack, stating, "even surrogate models that are too weak for direct transfer attacks can be used in our framework". Yan et al. (2021), also a Boundary Attack variant, attempts to learn to mimic the gradients of a victim net using a customised pre-trained policy network. Huang & Zhang (2020) learns a latent space of transferable adversarial features from its surrogate. Andriushchenko et al. (2020) supplies a bespoke feature bank which essentially transfers empirical domain knowledge (i.e. features determined to be likely to fool CNNs), both in the initialisation (vertical stripes) and the feature choice (homogeneous squares in the $\ell_\infty$ case, pairs of opposite-signed "decaying peaks" in the $\ell_2$). Sahu et al. (2020) attempts to perform a "black-box FGSM" by learning correlations between components of the loss function gradient within a Gaussian Markov random field framework. The "gradient priors" of Ilyas et al. (2019a) essentially amount to momentum and downsampling, and do not represent the sort of "flexible gradient transfer" we are otherwise discussing here.

**Optimisation algorithms:** Again, a standard optimisation framework for black-box score-based attacks is ascent on gradients estimated via numerical derivatives in guessed directions, possibly using enhancements such as priors and/or simplifications such as coordinate ascent. We consider the aforementioned SimBA variants (including the method we present) to essentially be of this type. Variations on or alternatives to this approach have included bandits (Ilyas et al., 2019a; Guo et al., 2019b), Natural Evolution Strategies (NES) (Li et al., 2019; Huang & Zhang, 2020; Dolatabadi et al., 2020) evolutionary algorithms (Liu et al., 2020; Wang et al., 2021), Bayesian optimisation (Ru et al., 2019; Shukla et al., 2019), and training of a policy network by REINFORCE (Yan et al., 2021). Al-Dujaili & O'Reilly (2020) uses a custom "flip/revert" approach based on checking the overall effect of grouped sign changes to pixel perturbations. Assuming submodularity, Moon et al. (2019) uses the max-heap-based "local search algorithm" to alternate between greedily inserting and removing elements defining a partition between oppositely signed perturbation pixels. Shi et al. (2019) comprises a set of heuristics meant to reduce the norms of attacks based on transfer from surrogate models: many of these are in principle applicable to attack vectors obtained otherwise.

## 5 CONCLUSION

We have demonstrated that score-based attacks using surrogates are in fact easy. By "easy", we mean that a very high success rate is achievable within a very low number of queries to the victim model, using an algorithm, GFCS, that is simple to both describe and implement. Further, this algorithm is based on a fairly direct type of transfer: it relies first and foremost on transfer of loss gradient directions from the surrogate to the target, falling back to transferring other combinations of features locally significant to the surrogate. One implication of this is that all of the examined surrogate and target networks are in fact similar to one another in the sense that the algorithm not only assumes, but entirely relies on. An interesting avenue for future work is to determine whether there are noteworthy situations in which the assumption of good feature transfer no longer holds, and the attack no longer suitable: this will be tantamount to identifying classes of networks that genuinely learn distinct responses from one another.

ETHICS STATEMENT

Our work introduces a novel (and very effective) black-box attack. As with all adversarial attacks, the method exposes a way of getting a target/victim network to issue responses to inputs that are almost certainly unlike those intended by its designer. Therefore, were an actual harmful practical application of such an attack to be identified in which the assumptions of this method (i.e. that the victim exposes its decision scores, and that a viable surrogate can be found) were satisfied, the publication of this method could in principle facilitate it. As is it not currently well understood how to build useful networks that do not have the sorts of properties being exploited here, there is no straightforward "defence". However, the primary perspective from which we approach this work is that of network analysis. A key goal is to make practitioners aware of potential risks their machine learning systems are exposed to, especially and most realistically *unintended* failure, and to motivate the community to design networks that are more robust, predictable, and comprehensible.

REPRODUCIBILITY STATEMENT

**Code and data:** We accompany this submission with the code implementing the proposed GFCS method. The code includes, in a utility file, lists of IDs of the ImageNet validation images used in the experiments, to allow full reproduction. Note that Algorithm 1 itself presents a complete and implementable description of the algorithm. We are also publicly releasing all code needed to reproduce all of the results in our paper. This will include details of any changes made to the released code of competing methods to standardise the comparison, including the issues noted in Appendix A.2.

**Statistical significance:** As described in Sec. 3.1, we report uncertainty for the results of all the reported methods, for all of the experiments. Moreover, we double the size of the test set typically used in the literature (and keep it fixed across competing methods).

**Hyperparameters:** Our method contains a single tunable hyperparameter (the step length $\epsilon$). As discussed in Appendix A.4, we chose its value on a held-out set and kept it fixed across all of the experimental configurations. For the competing methods, we have used the provided source code (or provided our own implementation when none existed, as and when noted) and the default hyperparameters, alongside any hyperparameter choice we found to work better than the default(s). For instance (see Appendix A.4), for SimBA-ODS, we found that $\epsilon = 2.0$ is significantly better than $\epsilon = 0.2$, which is the default used in Tashiro et al. (2020).

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

## A  APPENDIX

### A.1  COIMAGES AND DIMENSION REDUCTION

The assumption of local linearity is ubiquitous in the study of deep networks, and particularly adversarial attacks on them. A simple way of explaining the upper bound on the dimension of the coimage of a network's linear approximation is through the rank-nullity theorem. This points out that for linear transformation $T$ mapping finite-dimensional vector space $\mathcal{V}$ to vector space $\mathcal{W}$, $\text{rank}(T) + \text{nullity}(T) = \dim(\mathcal{V})$, where $\text{rank}(T) := \dim(\text{image}(T))$ and $\text{nullity}(T) := \dim(\text{kernel}(T))$. That is to say, the dimension of the input subspace that has any effect on the output ($\text{rank}(T)$) is no greater than that of the output space (because $\dim(\text{image}(T)) \leq \dim(\mathcal{W})$), and the dimension of the input subspace that has *no* effect on the output ($\text{nullity}(T)$) is at least equal to the difference between the input and output dimensions (because $\text{nullity}(T) = \dim(\mathcal{V}) - \text{rank}(T) \geq \dim(\mathcal{V}) - \dim(\mathcal{W})$). To make this more concrete, in the case of a standard ImageNet Inception-v3 network, $\mathcal{V} = \mathcal{R}^{299*299*3}$ and $\mathcal{W} = \mathcal{R}^{1000}$, and so in the linear approximation, $\text{rank}(T) = \dim(\text{coimage}(T)) \leq 1000$, and $\text{nullity}(T) \geq 267203$. Under the assumption that the features to which a surrogate is locally sensitive will largely transfer to a target network, naïve search of the input space is thus extremely wasteful.

This is the key issue which limits the preprint of Ma et al. (2020), which contains some similarities to our method in its preference for surrogate gradient transfer within a SimBA-like optimisation context. The crucial difference lies in their attempt to use standard RGF gradient estimation in order to progress optimisation when surrogate loss gradients fail, thus falling prey to the inherent issue with estimating gradients in high-dimensional input spaces pointed out in this context as far back as Chen et al. (2017). The dramatic difference in result quality between the two methods owes to this fact.

### A.2  IMPLEMENTATION DETAILS

As explicitly written in Alg. 1 and discussed in Sec. 2.2, the SimBA-style search is done within a standard PGA projection of the update candidate onto the feasible set, to avoid the common evaluation issue in the literature in which SimBA is punished *ex post facto* for violating a norm bound that the method was not originally designed to account for. Our method thus avoids this issue by design, but we further note that none of the evaluations we perform on other methods involve this sort of practice: if a bound is to be imposed, a method should always be modified in a straightforward manner to account for it, rather than being deemed to have failed after the fact.

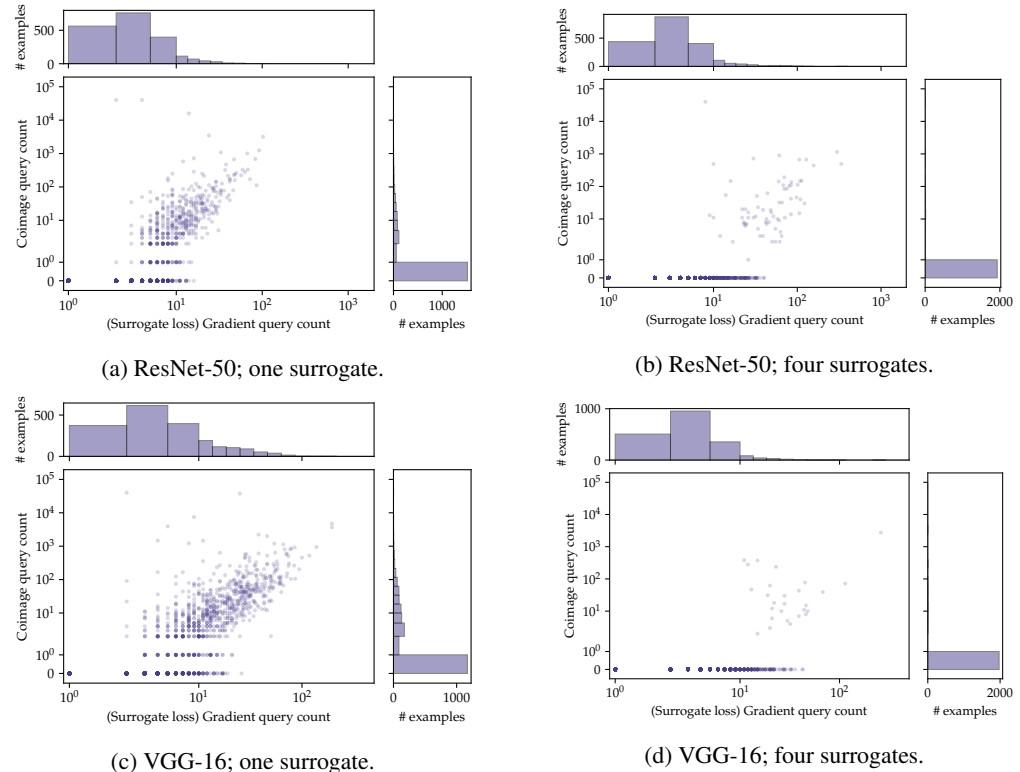

(a) ResNet-50; one surrogate.

(b) ResNet-50; four surrogates.

(c) VGG-16; one surrogate.

(d) VGG-16; four surrogates.

Figure 6: Breakdown of the total query count for the proposed method, *GFCS: Gradient First, Coimage Second*. The x-axis represents the number of queries required for a successful attack by the *gradient* part of the method, while the y-axis represents the number required by the *coimage* part. Histograms on the top and right sides of the scatter plots represent marginal empirical distributions. These plots correspond to the results depicted in Fig. 2.

On the other hand, our study of the implementations of competing methods has shown that some of them unnecessarily hobble their own performance by using surrogates with domains that differ from that of the target model without appropriately including interpolation between the domains. This typically shows up when Inception-v3, which is trained to accept an input in $\mathcal{R}^{299*299*3}$, is attacked using networks with the more common input domain of $\mathcal{R}^{224*224*3}$: adaptive pooling layers prevent crashing, but do not fix the issue of feature scale mismatch. As such, each surrogate in our method should be considered to contain an differentiable bilinear interpolation module as an initial layer, thus producing appropriately mapped gradients in the target domain on backpropagation. In the spirit of fair comparison, we implement this option for competitors as well. Additionally, as of this writing, the public LeBA implementation contains non-standard pre-processing of the input images: we replace this with standard practice, again to facilitate fair comparison against other methods.

Finally, note that when $L_{\mathbf{f}}(\mathbf{x})$ is the margin loss, its normalised gradient is just a special case of $\mathbf{d}_{\text{ODS}}(\mathbf{x}, \mathbf{f}, \mathbf{w})$ in which $\mathbf{w}_{c_t} \leftarrow 1$, $\mathbf{w}_{c_s} \leftarrow -1$, and $\mathbf{w}_{c \notin \{c_s, c_t\}} \leftarrow 0$. In our experimental setup, all methods are run on sets of images that, while otherwise randomly selected from the ILSVRC2012 validation set, are guaranteed to be correctly classified by each target network. Thus, there is never any difference between $c_s$ and the ground-truth label in the untargeted attack case.

## A.3 ADDITIONAL RESULTS FOR SECTION 3.2

The scatter plots and empirical marginal distributions in Fig. 6 complete the results of Sec. 3.2 and Fig. 3 from the main paper, extending the same analysis to ResNet-50 and Inception-v3 architectures. A very similar trend can be observed: 1) most of the examples are successfully attacked with a handful of queries, 2) the low count is largely due to the surrogate gradient transfer, and 3) using four surrogates amplifies the two previous phenomena.

## A.4 On the step length $\epsilon$

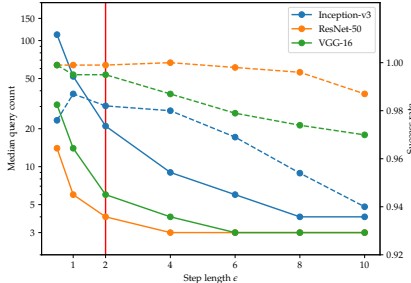

Figure 7: The figure illustrates two sets of curves. The solid curves represent the median query count required to successfully attack the three architectures as a function of the step length, while the dashed curves show the success rate.

The simple method we propose in Algorithm 1 accepts a single hyperparameter: the step length $\epsilon$ (line 14), which indicates the length of the perturbations to be attempted at each iterate along its candidate direction. We chose 2.0 as the default value for our experiments by performing a small grid search over a held-out set (disjoint from the 2000 examples used in the experiments of the main paper). Results for the three architectures used as victim (and ResNet-152 as the single surrogate) are shown in Fig. 7. Note that the figure has two $y$-axes and two sets of curves: solid and dashed curves should be considered looking at each axis on the left and right hand side of the plot, respectively. Clearly, better performance can be obtained by separately tuning this hyperparameter for each architecture. Instead, we simply chose a fixed value that is reasonable for all models, balancing the query count and success rate. Note also the fact that it is possible to achieve an even lower query count if one is willing to sacrifice about 1% of the success rate.

We observed that this choice of $\epsilon$ is also significantly better than the default one made by SimBA-ODS (Tashiro et al., 2020) (i.e. 0.2), and have reported results for both in our experiments.

## A.5 Experiments with $\ell_2$ norm bound $\nu = 5.0$

We repeat the experiments of Sec. 3.1 (whose results are displayed in Fig. 2), but instead using the norm bound of $\nu = 5.0$ sometimes used in $\ell_2$ adversarial attack experiments on ImageNet networks with input domains of size [224, 224, 3], such as VGG-16 and ResNet-50. This is roughly a factor of 2 smaller than the bound of $\nu = \sqrt{0.001D}$ used by our competitors and ourselves in the main experiments. The results of this experiment are given in Fig. 8. We see, as before, that GFCS comfortably dominates the low-query regime in all cases. In the single-surrogate cases, both GFCS and SimBA-ODS saturate at a slightly lower success rate than some competitors, including ODS-RGF. As already pointed out in Sec. 3.1, GFCS uses SimBA-ODS as its coimage sampler for the sake of simplicity, and inherits a small number of failure cases from it in some circumstances. This can easily be alleviated in exchange for a bit of additional complexity in the algorithm's implementation by using ODS-RGF to do coimage sampling.

## A.6 Experiments on CIFAR-10

We again repeat the main untargeted attack experiment, this time using CIFAR-10 as the dataset, with the target and surrogate networks sourced from https://github.com/akamaster/pytorch_resnet_cifar10. We use ResNet-110 as the target, and the much simpler ResNet-20 as its single surrogate. The $\ell_2$ norm bound is again set to $\nu = \sqrt{0.001D}$, which for CIFAR-10 $\approx 1.75$. Fig. 9 demonstrates the results. All methods completely solve the problem well within the 10K query limit: we focus the plot on the first 500 queries. This problem is easier than the main ImageNet version studied in this work, but GFCS nonetheless retains its relative dominance.

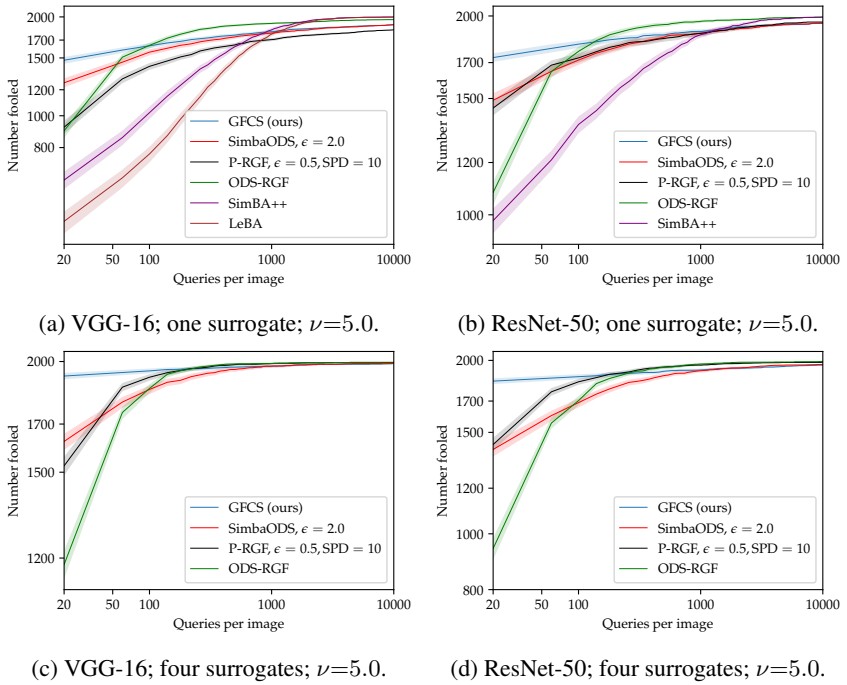

(a) VGG-16; one surrogate; $\nu$=5.0.

(b) ResNet-50; one surrogate; $\nu$=5.0.

(c) VGG-16; four surrogates; $\nu$=5.0.

(d) ResNet-50; four surrogates; $\nu$=5.0.

Figure 8: CDFs representing the number of successfully attacked examples at different query counts when performing untargeted black-box attacks on VGG-16 and ResNet-50, with one or four surrogates, under $\ell_2$ norm bound $\nu = 5.0$.

## A.7 ATTACKING A HARDENED NETWORK

As a final additional experiment, we repeat the four-surrogate untargeted attack of Sec. 3.1, using the adversarially trained Inception-v3 of Kurakin et al. (2017) (as provided by `https://github.com/rwightman/pytorch-image-models`) as the victim. The test set in this case consists of 800 ImageNet images correctly classified by the victim model. This model has been hardened on single-step adversarial attacks, and, as acknowledged by Kurakin et al. (2017) in their original presentation of it, is vulnerable to iterative white-box attacks. As we demonstrate in Fig. 10, it is also vulnerable to well-designed score-based black-box surrogate attacks, with GFCS again proving the most efficient of the attack methods.

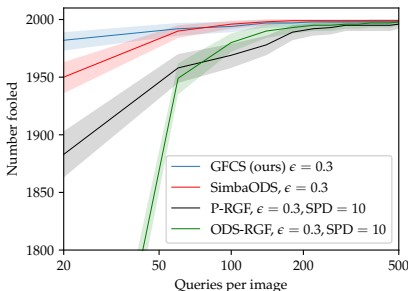

Figure 9: CDFs representing the number of successfully attacked examples (from **CIFAR-10**) at different query counts when performing untargeted black-box attacks on a ResNet-110 network using a ResNet-20 surrogate.

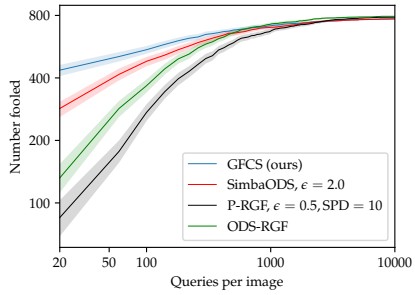

Figure 10: CDFs representing the number of successfully attacked examples at different query counts when performing untargeted attacks on an adversarially trained Inception-v3, using four surrogates. The target set in this case consists of 800 ImageNet images correctly classified by the adversarially trained victim network.

## A.8 BREAKDOWN ANALYSIS FOR TARGETED ATTACKS

Fig. 11 depicts the query breakdown for the targeted attack experiment of Sec. 3.4. Fig. 5 has already demonstrated that GFCS outperforms its competitors dramatically on this problem: the breakdown sheds light on how it is able to solve this difficult problem efficiently. The trend is broadly similar to that in the analogous plots for the untargeted attacks: a large number of examples are solved entirely using loss gradient transfer, while the others rely on the backup at least some of the time. What is remarkable about this plot relative to those of the untargeted attacks is that the large number of loss-gradient-only examples are a large *minority*: more of the 1000 attacked images require some use of coimage sampling than do not, in this case. The loss-gradient transfer is crucial to the considerable outperformance of GFCS over SimBA-ODS seen in Fig. 5, while the use of SimBA-ODS as a backup is critical in preventing the overall method from failing the majority of the time.

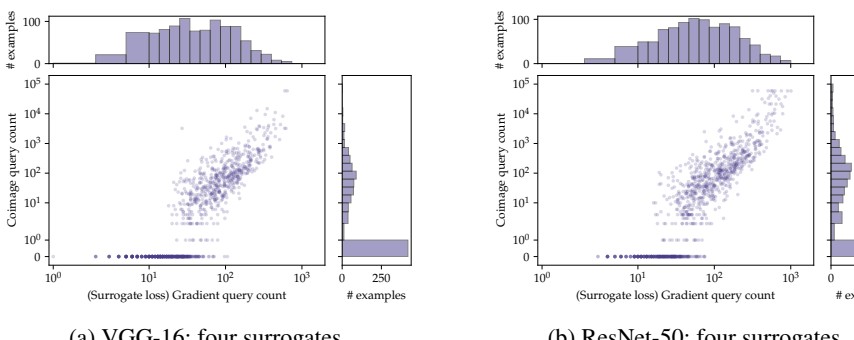

(a) VGG-16; four surrogates.  (b) ResNet-50; four surrogates.

Figure 11: Breakdown of the total query count for GFCS, as in Figs. 3 and 6, for the targeted attack experiments whose results are depicted in Fig. 5. See the caption of Fig. 3 if a reminder of what the figure depicts is required.

