# OpenReview forum: "Attacking deep networks with surrogate-based adversarial black-box methods is easy"
_ICLR.cc/2022/Conference — ICLR 2022 Poster_

### Official Review · Reviewer_KEn1 · 2021-10-25

**Correctness:** 4
**Technical Novelty And Significance:** 2
**Empirical Novelty And Significance:** 2
**Recommendation:** 6
**Confidence:** 4

**Main Review:**

[Pros]:
* The paper is well-motivated and easy to follow.
* The proposed algorithm is simple but achieves state-of-the-art performance.

[Cons]:
* The technical contribution is marginal.
* The empirical comparison is insufficient.
* The result and discussion in Sec. 4.4 is digressed from the main statement.

The paper proposes a simple score-based black-box attack algorithm which achieves state-of-the art results. The simplicity and  effectiveness of the proposed algorithm is great for the research community. However, it seems that the novelty is technically marginal. Moreover, the empirical comparisons are not sufficient to show the advantage of the proposed method.

* The proposed algorithm is a natural combination of two existing approaches. A previous study [1] utilized the surrogate’s loss gradient for improving SimBA. On the other hand, SimBA-ODS utilized the coimage of the surrogate’s local linearization for improving SimBA. The proposed algorithm is a simple switch algorithm of the above methods based on the update success, and thus the technical contribution is marginal.

* The empirical comparison is insufficient, because the proposed algorithm was not compared with a variant, which only uses the surrogate’s loss gradient without the coimage of the surrogate’s local linearization (i.e., similar to [1]). Since the proposed algorithm is the combination of two existing approaches (i.e., gradient-based and coimage-based), the algorithm should be compared with both approaches. The right panel in Figure 3 implies that the variant will achieve very similar query efficiency with similar success rate, so the lack of the comparison is problematic. In addition, Figure 3 also suggests that the experiment setting in Sec. 4.1 and 4.2 is suitable for methods utilizing the surrogate’s loss gradient. To show the advantage of the combination of both existing approaches, the evaluation in more difficult settings (such as under small norm bound and on adversarially-trained robust models) is also important. Although the evaluation of targeted attacks in Sec. 4.3 is one of the difficult settings, the comparison and discussion only focus on query efficiency and do not explicitly shed light on the effect of the combination like Figure 3.

* The result in Sec. 4.4 is interesting, but it is not directly related to GFCS. Previous studies have already shown the importance of local gradient information, and GFCS combines two approaches which utilize local gradient information in a different way. It seems that Sec. 4.4 just focuses on input-specific priors and using surrogates, which are related to all local gradient-based methods and not specific to GFCS. How does Sec. 4.4 connect with the main contribution of GFCS?

Minor comments
* the experiment setting for SimBA-PCA-gradients is not clear. What is a sample input set? Is it different from the test dataset? How many adversarial examples are generated per image?
* SimBA-SVD is sometimes miswritten as SimBA-PCA.
* In the page 9, what is PGA?

[1] Jinghui Cai, Boyang Wang, Xiangfeng Wang, and Bo Jin. Accelerate Black-Box Attack with White-Box Prior Knowledge. IScIDE 2019.

-----Post rebuttal comments-----

After reading all rebuttals, I'm slightly inclined to accept, since most of my concerns are addressed.

While I raise the score, I'm still concerned about the structure and writing in terms of Sec 4.4. I don't think the current paper properly conveys the author's claim. For example, the first sentence in Sec 4.4 does not distinguish GFCS from other methods, so It seems that GFCS can be replaced with other methods. The authors also do not define "local information". The connection between Sec 4.4 and other parts is weak in the current paper, so the authors should improve the structure and writing when the paper is accepted.

**Summary Of The Paper:**

The paper proposes a new simple black-box adversarial attack algorithm, which utilizes surrogate models. The algorithm attempts iterative ascent in the direction of the surrogate’s loss gradient, and when the attempt fails, the algorithm searches alternative directions by exploiting the surrogate’s Jacobian’s row space. The experiments show that the proposed algorithm produces very low query counts with low failure rates, compared to existing baselines.

**Summary Of The Review:**

While the proposed black-box attacks show strong empirical performance, the technical contribution is marginal. Moreover, the empirical comparison is not enough to show the advantage of the proposed method.

---

> ### Author Response · Authors · 2021-11-16
> **Please see "re: The Requests for Comparisons Against Gradient-Only Methods"**
>
> > "The empirical comparison is insufficient, because the proposed algorithm was not compared with a variant, which only uses the surrogate’s loss gradient without the coimage of the surrogate’s local linearization (i.e., similar to [1]). Since the proposed algorithm is the combination of two existing approaches (i.e., gradient-based and coimage-based), the algorithm should be compared with both approaches. The right panel in Figure 3 implies that the variant will achieve very similar query efficiency with similar success rate, so the lack of the comparison is problematic. In addition, Figure 3 also suggests that the experiment setting in Sec. 4.1 and 4.2 is suitable for methods utilizing the surrogate’s loss gradient."
>
> Thank you for your review. We will be responding to your other points shortly, but first, as you have made a request that is similar to that of other reviewers, we have written a joint response to all of you. Please see https://openreview.net/forum?id=Zf4ZdI4OQPV&noteId=Cu2Gp36dfNz .

---

> ### Author Response · Authors · 2021-11-17
> **On the Method's Simplicity**
>
> > "Pros: The paper is well-motivated and easy to follow. The proposed algorithm is simple but achieves state-of-the-art performance... Cons: The technical contribution is marginal."
>
> > "The simplicity and effectiveness of the proposed algorithm is great for the research community. However, it seems that the novelty is technically marginal."
>
> > "The proposed algorithm is a simple switch algorithm of the above methods based on the update success, and thus the technical contribution is marginal."
>
> We thoroughly agree that this is a very simple method, and actually believe this to be one of the main strengths of our work. However, the method’s simplicity seems to have worked both for and against us. What if we had presented a more complicated method that achieved results better than those of the prior art, but slightly worse than the ones we've presented here? Would that have constituted a greater or lesser technical contribution?
>
> Further, there is *no* realistic situation that we know of in which the competitors’ assumptions hold but ours do not, i.e. in which loss gradients seldom transfer and yet e.g. the coimages often do. The evidence thus far (from our runs of their experiments) indicates that this does not typically occur.
>
> Note that not every simple method that is superficially similar to ours will work: we have given an example of this fact in Appendix A\.1. A separate but related comment applies to methods that attempted to rely entirely on direct gradient transfer: had they adopted our perspective, they could have implemented a well-designed switching method of this sort, which would have alleviated their failure cases (see the previous response). We were not able to find an example of any paper which successfully combined all of the relevant points to demonstrate the results that we have. This, we believe, is our main technical contribution.

---

> > ### Comment · Reviewer_KEn1 · 2021-11-19
> > **Re: On the Method's Simplicity**
> >
> > While I feel the proposed method is a natural variant of ensemble attacks (e.g., using multiple surrogates), I have found the combination of different updated methods for black-box attacks has not been well investigated. I now agree there are some technical contributions.

---

> ### Author Response · Authors · 2021-11-17
> **The Relevance of Section 4.4**
>
> > "The result and discussion in Sec. 4.4 is digressed from the main statement."
>
> > "The result in Sec. 4.4 is interesting, but it is not directly related to GFCS... It seems that Sec. 4.4 just focuses on input-specific priors and using surrogates, which are related to all local gradient-based methods and not specific to GFCS. How does Sec. 4.4 connect with the main contribution of GFCS?"
>
> We agree that this is something of a digression, though one we considered worthwhile. We’re glad the reviewer also found the results interesting. Perhaps we could do a better job of contextualising the section, as follows:
>
> The paper, up until Sec. 4.4, has clearly established that there is a large amount of target-network behaviour effectively encoded by the surrogate networks, which can thus be used to craft transfer-based adversarial attacks very efficiently. This involves storing and differentiating the surrogate networks in full. An interesting question arises: could the representation be simplified further by distilling the surrogate into a collection of basis vectors representing an adversarial subspace which would then be searched online? That is, can we throw away all of the location-dependent information contained in the surrogate’s parameters and leave it to the optimiser to infer the relevant example-specific local information? Can the transfer be made even *simpler*?
>
> Frankly, this was actually an earlier hypothesis of ours, and of [Guo et al. 2019a] (the original SimBA work), who wrote in their conclusion, “One possible extension could be to further investigate the selection of different sets of orthonormal bases, which could be crucial to the efficiency of our method by increasing the probability of finding a direction of large change.” After all, as discussed in Sec. 4.4, it has long been established that adversarial subspaces and universal adversarial perturbations exist, making the hypothesis reasonably credible: these are location-agnostic phenomena, by definition.
>
> In fact, we can informally report here (based on unpublished SimBA-SVD experiments) that a nontrivial amount of location independence *can* be exploited in CIFAR10 networks in exactly this manner, *but*, the results do not carry through to ImageNet. We found the latter negative result to actually be more significant than the former "positive" one, and chose to focus on it instead, which is what the experiments of Sec. 4.4 represent. Further, we consider it to be further evidence that the behaviour on “real” problems should not be extrapolated from behaviour on datasets such as CIFAR10: see e.g. the final slide from the Kolter section of the NeurIPS 2018 tutorial on adversarial examples for an example of someone else making a closely related point regarding robustness on MNIST and the fact that such phenomena can be dataset-specific, especially when the dataset is simplistic: https://adversarial-ml-tutorial.org/adversarial_ml_slides_parts_2_3.pdf
>
> So, GFCS illustrates that surrogates typically encode crucial information about their victims, while Sec. 4.4 indicates that there are limits to how much further that information can be reduced while retaining the demonstrated level of effectiveness, at least within this optimisation context. The surrogates appear to be fairly parsimonious for the task at hand, and the hypothesis suggested at the end of [Guo et al. 2019a] appears to be rejected. The surrogates are indeed providing crucial location-specific information at each point, notwithstanding the existence of universal adversarial perturbations, transferable adversarial subspaces, etc.
>
> > “Previous studies have already shown the importance of local gradient information, and GFCS combines two approaches which utilize local gradient information in a different way.”
>
> Could the reviewer provide the references in mind, regarding the demonstration of the importance of local gradient information? If so, we would be happy to incorporate those and discuss them in this context. We do also believe that the literature has established the importance of local information in some senses (e.g. DeepFool perturbations are generally much smaller than the universal adversarial perturbations derived from them), but, as above, we thought there were interesting open questions around this matter relevant to the paper’s larger point.

---

> > ### Comment · Reviewer_KEn1 · 2021-11-19
> > **Re: The Relevance of Section 4.4**
> >
> > Thank you for the detailed response. However, I still think the topic in Sec 4.4 interrupts the flow of the paper. I explain my concerns in detail as follows.
> >
> > GFCS shows that both gradients and coimages are important for query-efficient attacks, and this is the main statement in the paper. However, Sec 4.4 does not mention this point. In addition, the topic in Sec 4.4 can be discussed for other attack methods which utilize local information such as SimBA-ODS. In other words, Sec 4.4 is not related to the characteristic specific to GFCS. Thus, I feel there are two different stories in a paper and Sec 4.4 interrupts the flow of the paper.
> >
> > Moreover, the comparison between GFCS and the other methods in Sec 4.4 will not explicitly show the importance of local information itself, but show the importance of both gradients and coimages to some extent. Actually, if GFCS is replaced with "GFCS - ODS" in Sec 4.4, "GFCS - ODS" will not outperform the other methods, though "GFCS - ODS" utilizes local information.
> >
> > >Could the reviewer provide the references in mind, regarding the demonstration of the importance of local gradient information?
> >
> > What I mean is that there are existing attacks utilizing local information such as SimBA-ODS and P-RGF, as I mentioned above.

---

> > > ### Author Response · Authors · 2021-11-23
> > > **Re:**
> > >
> > > In a nutshell: The paper up until Sec. 4.4 establishes that local information from surrogates is *sufficient* to solve the adversarial attack problem. Sec. 4.4 establishes that local information is also *necessary* for that level of performance. We view the sufficiency and necessity as a point worth making jointly, even though we do agree that it makes the paper less simple than focusing entirely on the sufficiency argument.
> > >
> > > First, let us note that there is more to "local information" than the loss function gradient. The local first-order approximation of a function comprises its value (the output score vector) and its derivative (the score Jacobian, whose row space is the coimage) at the given location. The loss gradient is just a specific vector in the coimage (which is often, but not always, useful). The fact that "GFCS-ODS" often fails doesn’t prove anything about the possibility of leveraging local information from the surrogate, because it discards nearly all of that information. Other methods, particularly GFCS, demonstrate that this information can be used to achieve excellent performance on this problem, if used properly. That part is not in question.
> > >
> > > However, none of those methods, including GFCS itself, represent proof that this sort of local information is *necessary* for such a level of performance. Someone might claim that a method obeying stricter assumptions, such as having to form its feature set entirely offline, could deliver comparable performance. In fact, Reviewer SLVp has asked us for precisely such comparisons (against Square Attack), so this is not a trivial or hypothetical point.
> > >
> > > What 4.4 demonstrates is that, at least within the SimBA optimisation context and under certain reasonable assumptions, *no* method relying on a location-agnostic offline feature set will *ever* manage to surpass a particular level of performance which falls well short of GFCS. The choice of SVD on adversarial directions collected over a training set is not arbitrary: it is in a sense the optimal L2 attack basis for this optimisation method if it is assumed that the distribution of adversaries on the test set matches that of the training set. We have now supplemented this point by also providing the comparison against L2 Square Attack requested by SVLp. Just as the Bandits paper [Ilyas et al., 2018] argued that no method which did not leverage a prior of some sort could ever hope to surpass RGF estimation, we are arguing that empirical evidence indicates that no method of this type whose prior is not itself a function of image space will ever be able to compete with GFCS on these evaluation metrics.

---

> ### Author Response · Authors · 2021-11-18
> **re: Minor Comments**
>
> > "the experiment setting for SimBA-PCA-gradients is not clear. What is a sample input set? Is it different from the test dataset? How many adversarial examples are generated per image?"
>
> Good question. The sample input set in this case is a random subset of the ImageNet training set, and the evaluation is done on a selection from the ImageNet validation set. One sample adversary is generated per sample image, using the method described in footnote 2 in Sec. 4.4 (the gradient of a randomly selected class score).
>
> In principle, as many sample images/adversaries can be used to construct the data matrix as one likes (including multiple class-score gradients per image), and a larger sample set could improve performance by better representing the population gradient statistics. However, in our experiments, we found that this performance saturated surprisingly early w.r.t. the sample count, and there wasn’t any evident advantage over using a minimal set: 10K examples to define a 10K-D SimBA basis, which is more than enough for a 10K query limit when each direction will consume one or two queries. Note that if the principal component matrix is higher-dimensional, the method will still only be able to utilise the first 10K (or fewer) directions due to the query limit.
>
> > "SimBA-SVD is sometimes miswritten as SimBA-PCA."
>
> Indeed, thank you for catching that. (We internally referred to these interchangeably because of the direct relationship between PCA and the SVD: writing both versions in the paper was an accident.) We have replaced all instances of "SimBA-SVD" with "SimBA-PCA" (as it's called in the graph legend).
>
> > "In the page 9, what is PGA?"
>
> Projected Gradient Ascent. The abbreviation is defined on page 4. We have used ascent sign conventions throughout the paper, i.e. the attacker seeks to increase the loss. Thus, we refer to PGA rather than PGD.

---

> ### Author Response · Authors · 2021-11-23
> **re: Evaluation in More Difficult Settings**
>
> re: adversarially trained networks and smaller norm bounds (among other new experiments we have included in the revised draft), please see the response to SLVp here: https://openreview.net/forum?id=Zf4ZdI4OQPV&noteId=yVc-dmUHnDw

---

### Official Review · Reviewer_tmFJ · 2021-10-31

**Correctness:** 4
**Technical Novelty And Significance:** 2
**Empirical Novelty And Significance:** 2
**Recommendation:** 6
**Confidence:** 2

**Main Review:**

First of all, I should admit that I am no expert in this area and hence my assessment may be affected by the lack of expertise.

Language-wise the paper seems to be well written. The discussion flow is clear and easy to follow although there are a number of concepts that I am personally not familiar with, and so I believe it would help to define all the notation and the concepts studied (besides the basic definition of an adversarial attack) for a reader like me.

The algorithm proposed is well motivated and described. Related work seems to be properly analysed. Furthermore, although I am not familiar with the standard experimental setup in this area, I should say the results look good to me and the advantage of the proposed algorithm over the state of the art looks convincing.

My understanding is that despite the demonstrated practical efficiency of the proposed solution, it is not guaranteed to successfully find an adversarial example even if such an example exists, due to the limitations of the methods used. I wonder if the authors tried to compare their approach to the approaches based on formal methods for reasoning about the model behaviour. Also, it would help to mention this line of work in the related work.

**Summary Of The Paper:**

This paper is devoted to the problem of generating adversarial examples to the predictions of machine learning models. That is, given an instance predicted by a target ML model one way, the paper aims at detecting another instance that is relatively close to the given one but such that the prediction of the model for the new instance is different. The paper proposes a novel adversarial attack algorithm, which behaves as twofold: (1) it tries to apply iterative ascent in the direction of the surrogate's loss gradient until it can and then (2) samples alternative directions from the coimage of the surrogate's local linearisation. The new approach is claimed to be a fast and practical attack as it significantly outperforms the competition in terms of the number of queries needed to generate adversarial examples successfully.

**Summary Of The Review:**

Overall, I find the paper to offer a solid contribution that may be interesting enough for the community working in the area of black-box adversarial attacks, if accepted to ICLR.

---

> ### Author Response · Authors · 2021-11-17
> **re: Formal Methods**
>
> > "I wonder if the authors tried to compare their approach to the approaches based on formal methods for reasoning about the model behaviour."
>
> Thank you for your review. Formal methods, unfortunately, do not scale to problems of the scope being considered here. The field is limited to search-based strategies.

---

### Official Review · Reviewer_wR15 · 2021-11-02

**Correctness:** 2
**Technical Novelty And Significance:** 2
**Empirical Novelty And Significance:** 2
**Recommendation:** 5
**Confidence:** 3

**Main Review:**

The success of the proposed idea is surprising, while this effect is not explained by the experiments in the paper.
In particular, all attacks considered by the authors as competitors (SimBA++, SimbaODS, LeBA, P-RGF, ODS-RGF) directly assume, that surrogates are not precise enough to replace an initial model.
So, the evaluation is incomplete. if we don't consider benchmarks that use gradient steps that are taken directly from surrogate models.
As the authors demonstrate in Figure 3 we require only a small number of coimage steps, while most of the steps are taken directly from a surrogate model (or surrogate models).

The authors provide some explanation in Section 4.4 on why do we need local gradient information.
Common sense and previous work they cite (e.g. P-RGF) also make similar statements.
It would be interesting to consider the correlation between used surrogates and the victim model and quality of attack, as we can easly degrade the quality of a surrogate or maximize discrepancy between a surrogate model and a victim model by incorporating a specific term in the loss function.

The idea of ensembles for the performance improvement seems interesting (also we can try to apply something similar to bandits to select the best surrogate during the optimization?).

**Summary Of The Paper:**

The paper proposes a simple, yet effective adversarial attack in a black box scenario with available surrogates. The attack combines a classic gradient attack via an ensemble of surrogates and an ODS attack.

**Summary Of The Review:**

The authors suggest an interesting effect but provide a little investigation of it.
The cause of this effect can be some specifics of used pairs surrogate-victim or more general effect related to the correlation of deep learning models trained on large datasets and on general transferability.

As I don't get from the paper, why this phenomenon occurs, I suggest starting a discussion to find out, if it is possible to improve the paper in this aspect. Now, this is the obstacle, that makes the paper below the acceptance threshold.

---

> ### Author Response · Authors · 2021-11-16
> **Please see "re: The Requests for Comparisons Against Gradient-Only Methods"**
>
> > "So, the evaluation is incomplete. if we don't consider benchmarks that use gradient steps that are taken directly from surrogate models. As the authors demonstrate in Figure 3 we require only a small number of coimage steps, while most of the steps are taken directly from a surrogate model (or surrogate models)."
>
> Thank you for your review. We will be responding to your other points shortly, but first, as you have made a request that is similar to that of other reviewers, we have written a joint response to all of you. Please see https://openreview.net/forum?id=Zf4ZdI4OQPV&noteId=Cu2Gp36dfNz .

---

> ### Author Response · Authors · 2021-11-17
> **On (1) The Competitors' Assumptions vs. Reality; (2) The Design and Demonstration of Our Method; and (3) Surrogate Degradation**
>
> > “In particular, all attacks considered by the authors as competitors (SimBA++, SimbaODS, LeBA, P-RGF, ODS-RGF) directly assume that surrogates are not precise enough to replace an initial model."
>
> While our competitors do indeed state in their respective papers that their methods assume that surrogates give only imprecise information about the target model, the reality is that their main results were demonstrated on networks for which a much stronger form of transfer was actually available, as proven by our results on those exact networks. It's not that the surrogates poorly represent the target networks, it's that the competitors do not fully leverage the transferable gradient information stored in those surrogates. The competitors' failure to leverage this information as successfully as we have did not constitute evidence that those surrogates did not model the target networks very closely.
>
> Moreover, the competitors are not actually free of the transfer assumption themselves: ODS, for example, relies entirely on coimage transfer, though it does not state its assumption in this way. Like all of the competitors, it does benefit from the high similarity between the surrogates and targets, just not as efficiently as we do.
>
> We welcome further discussion of this point if there is any remaining misunderstanding or controversy.
>
> > "The success of the proposed idea is surprising, while this effect is not explained by the experiments in the paper.”
>
> Regarding the surprise at the success of our method: we based this design on the knowledge contained in the existing literature on transferable adversarial examples. Adversarial examples are typically identified through simple first-order optimisation methods, and the established fact of their transferability between networks indicated to us that a transfer-first method within a sensible optimisation framework would likely succeed.
>
> We argue that the effect is very much explained by the results in the paper: our method’s assumptions are stated very clearly, and, by construction, our method could not demonstrate the results that it does were these assumptions not largely satisfied. Thus, the answer is straightforward: our method works efficiently because all of these networks do in fact contain transferable gradient information. This indicates that the different nets respond very similarly to similar changes in the inputs, which is why even the targeted experiment works as well as it does. Figs. 3 and 6 make this fact very explicit by showing gradient usage on a per-example basis.
>
> We can certainly elaborate on this in the text, if we can come to an agreement about this with the reviewer. We would be happy to discuss these arguments further, on this forum.
>
> > "It would be interesting to consider the correlation between used surrogates and the victim model and quality of attack, as we can easily degrade the quality of a surrogate or maximize discrepancy between a surrogate model and a victim model by incorporating a specific term in the loss function."
>
> Although synthetically forcing networks to be dissimilar under some definition may be straightforward in practice, it does not have a straightforward interpretation. If, for example, analytical gradients were forced to have high cosine distance at corresponding sample points, this would likely just encourage the gradient masking/obfuscation examined in [Athalye et al.]. We agree with that work’s perspective that this represents a trivial annoyance for an attacker to bypass, not a meaningful change in the networks’ behaviour. In general, the subfield of adversarial training illustrates that it is not trivial to alter a network’s behaviour more fundamentally while maintaining its accuracy on the target task, even if it is trivial to design an objective function or augment the data.
>
> As we have written in the conclusion, finding a non-contrived case that violates our assumption is a matter for future work, for ourselves and the field in general.
>
> [Athalye et al.] *Athalye, Anish, Nicholas Carlini, and David Wagner. "Obfuscated gradients give a false sense of security: Circumventing defenses to adversarial examples." In International conference on machine learning, pp. 274-283. PMLR, 2018.*

---

> ### Author Response · Authors · 2021-11-23
> **Re: The Benefits of Local Information Being a Matter of Common Sense**
>
> Note that it is not obvious that local information is *required*, only that it is *useful*. Universal adversarial perturbations (as discussed in the paper) are an example of a location-agnostic adversarial phenomenon. See the detailed discussion with KEn1 beginning here: https://openreview.net/forum?id=Zf4ZdI4OQPV&noteId=b9ap86QVb5E

---

### Official Review · Reviewer_SLVp · 2021-11-02

**Correctness:** 3
**Technical Novelty And Significance:** 2
**Empirical Novelty And Significance:** 3
**Recommendation:** 5
**Confidence:** 4

**Main Review:**

Positives:
1. The proposed approach is simple, effective, and query-efficient.
2. The literature review seems quite thorough and does an especially nice job of covering the related work in this domain. It would be good to discuss some recent Bayesian optimization approaches [1, 2] to make it a complete review.
3. Experiments are well designed to show the effectiveness of the proposed framework on both untargeted and targeted attacks.
4. The paper is well written and easy to follow.

Concerns:
1. Although the experiments are well-designed, the authors use just one dataset to show the effectiveness of the approach. It would be interesting to see if the proposed approach achieves similar performance on other datasets e.g. CIFAR10.
2. It is hard to compare the performance of different approaches from Table 1 as they all are performing really well. For example, Simba-ODS performance is very close to that of GFCS. How is the value of epsilon been chosen as it significantly affects the performance of Simba-ODS? Is it possible to further tune this parameter to further improve the performance? It would be also interesting to see if the performance difference increase if you decrease the L2 norm bound to 5 which has been used in several prior work.
3. The authors are missing a comparison to the SOTA method for score-based attacks [3]. it would be also good to compare with Bayes-attack approaches [1, 2] as they show similar properties as the proposed approach, i.e., successfully attack a large fraction of examples
within a very low number of queries.
4. The authors are also missing a comparison to SOTA optimization-based methods for targeted attacks.
5. I have some concerns with the setting of the experiments. The surrogate models are trained on the same dataset and use the same loss function which is already a lot of information for the adversary to attack the target model. This setting should not be classified as the typical black-box setting.
6. Since the transfer attack using the surrogate models are easy for the setting considered here, it would be an interesting ablation to apply FGSM or PGD attack using the gradient of the surrogate model(s) and compare the attack success rates.
7. Based on the experiment in Section 4.2, the ODS is seldom used for untargeted attacks. The authors should provide ablation experiments such as GFCS - ODS to show the effect of ODS on the attack success rates.

References:
1. Ru, B., Cobb, Blaas, & Gal. (2019). BayesOpt Adversarial Attack. ICLR 2020.
2. Shukla, S. N., Sahu, A. K., Willmott, D., & Kolter, J. Z. (2019). Black-box adversarial attacks with bayesian optimization. arXiv preprint arXiv:1909.13857.
3. Maksym Andriushchenko, Francesco Croce, Nicolas Flam- marion, and Matthias Hein. Square attack: a query-efficient black-box adversarial attack via random search. In ECCV, 2020.

**Summary Of The Paper:**

This work presents an adversarial attack in the black-box setting where an adversary has access to only logit outputs from the target model. The proposed approach combines the ideas from transfer-based attacks and zeroth-order optimization methods for score-based attack methods. Experiments on multiple architectures show that the proposed approach is query efficient while achieving similar success rates compared to recent approaches.

**Summary Of The Review:**

Although there are several good things about the paper because of its simplicity and effectiveness, the authors should provide more experimental evidence to show the effectiveness of the proposed approach. There are also some concerns based on the setting of the experiment (pseudo-black-box setting).

---

> ### Author Response · Authors · 2021-11-16
> **Please see "re: The Requests for Comparisons Against Gradient-Only Methods"**
>
> > "... it would be an interesting ablation to apply FGSM or PGD attack using the gradient of the surrogate model(s) and compare the attack success rates... The authors should provide ablation experiments such as GFCS - ODS to show the effect of ODS on the attack success rates."
>
> Thank you for your review. We will be responding to your other points shortly, but first, as you have made a request that is similar to that of other reviewers, we have written a joint response to all of you. Please see https://openreview.net/forum?id=Zf4ZdI4OQPV&noteId=Cu2Gp36dfNz .

---

> > ### Comment · Reviewer_SLVp · 2021-11-29
> > **Response follow up**
> >
> > I would like to thank the authors for providing additional experimental evidence. There are still a few concerns so I have decided to keep my score. First, the additional results in Appendix (A.5, A.6, and A.7) show that other baseline approaches can achieve the same or better success rate than the proposed approach if the query budget is provided up to 1000 or more. The proposed approach saturates early and shows no further improvements when more queries are allowed. This makes me question the claim of the paper and whether this paper is only effective in smaller query budgets. Also, it's not clear how the approaches stack in terms of median/average counts in the new experiments. Second, for CIFAR experiments, I think the L2 bound selected is looser than other papers and that's why the problem seems further easy. I would recommend using an L2 bound of 0.5. Finally, I'm not convinced on the threat model and most of the experimental evidence shows that the research for this threat model is almost saturated (and the problem seems almost solved).

---

> > > ### Author Response · Authors · 2021-11-30
> > > **Re: Response Follow-Up**
> > >
> > > > "First, the additional results in Appendix (A.5, A.6, and A.7) show that other baseline approaches can achieve the same or better success rate than the proposed approach if the query budget is provided up to 1000 or more. The proposed approach saturates early and shows no further improvements when more queries are allowed. This makes me question the claim of the paper and whether this paper is only effective in smaller query budgets."
> > >
> > > We are somewhat surprised to hear that this is how the reviewer has interpreted the results of the requested additional experiments. When other methods achieve the same success rate but take more queries to do so, that is precisely the main point of the paper. "Success", in this context, is measured by our curves sitting above the other curves, as they generally do.
> > >
> > > In the cases where the success rate reached by the query limit is slightly smaller for GFCS (and SimBA-ODS) than that of some competitors (as in Fig. 8, which, as requested, used a smaller norm bound than was actually used in any of the competiting papers themselves), that is due to the limitations of using SimBA-ODS as the coimage sampler, as explained in the main paper:
> > >
> > > "Note that our choice of SimBA-ODS as the coimage sampler is partly about simplicity: as the results demonstrate, a very small number of failures are expected when it is used on its own, and we effectively inherit them. At the cost of a bit of added complexity in the implementation, e.g. ODS-RGF could be substituted, and would likely lead to further improvements in both the failure rates and query count curves, while still representing a form of GFCS."
> > >
> > > When saturation occurs, it is to the limit imposed by SimBA-ODS: the difference is that GFCS reaches this limit much faster, because it is much more efficient. Besides this, many would argue that it is the low-query-count regime that represents the most important and realistic setting, and the success rate gaps are in any case very small when they do occur, as can be seen through inspection.
> > >
> > > > "Also, it's not clear how the approaches stack in terms of median/average counts in the new experiments."
> > >
> > > We sometimes provide medians as summary statistics (as in Table 1) to aid the digestion of large amounts of information across competing methods. However, the chosen summary statistic is inherently arbitrary: the CDF curves are the main results and should be treated as such.
> > >
> > > > "Second, for CIFAR experiments, I think the L2 bound selected is looser than other papers and that's why the problem seems further easy. I would recommend using an L2 bound of 0.5."
> > >
> > > The CIFAR experiments are entirely additional, and involved modifying the competing methods to support CIFAR, as these experiments were not done in any of those papers. We again used the dimension-adaptive L2 norm bound of sqrt(0.001D), as that was what was used in those papers for their main experiments (and as no other value was requested in the original review). Note that all of the adversaries being discussed are visually tiny, and a factor of roughly 2 (which is what separates sqrt(0.001D) and 0.5 for CIFAR10) is negligible in the sense of perceptibility.
> > >
> > > > "Finally, I'm not convinced on the threat model and most of the experimental evidence shows that the research for this threat model is almost saturated (and the problem seems almost solved)."
> > >
> > > Regarding the threat model, we reiterate that this is a recently established branch of the adversarial attack literature, and we are making a point about that fact. We have discussed this here: https://openreview.net/forum?id=Zf4ZdI4OQPV&noteId=WKFo3Wmdle

---

> ### Author Response · Authors · 2021-11-16
> **On Concerns About the Threat Model**
>
> > "I have some concerns with the setting of the experiments. The surrogate models are trained on the same dataset and use the same loss function which is already a lot of information for the adversary to attack the target model. This setting should not be classified as the typical black-box setting."
>
> We actually quite agree that this setting should not be classed as the typical black-box setting! We view this threat model (score-based attacks using surrogates) to be something worthy of study and critique precisely because it has taken hold in the literature while managing to avoid the point of just how inherently easy the problem is as specified. Given that all of the competitors we identified were relatively modern NeurIPS papers, we felt that this required addressing. The attacker’s job is indeed far easier than it is in more restricted threat models, which is why we chose the title for the paper that we did. But in making that critique, we are using the terminology that the field has already adopted, referring to this as a specific type of black-box method (“surrogate-based”). We propose adding a clarifying sentence at the end of the introduction such as the following: “Given this fact and its implications for working with these networks under this threat model, we believe the community should carefully evaluate how and whether it wants to continue approaching this problem.”
>
> Regarding the experimental setup otherwise: as detailed in the paper, we deliberately allowed the prior art to dictate the terms of our own experiments. We've taken great pains to make sure that we used the same threat model, success measures, parameter values (where relevant/appropriate), networks, and data as the competitors. Note that this does indeed involve the surrogate networks being trained on the same dataset as the target.
>
> But having agreed on the easiness of the threat model (from the attacker’s perspective), we’d say that there is some genuine controversy about its *realism*. It is arguable that there is a limit to how dissimilar one should ever expect two deep nets trained on the same problem to be. The networks studied in the prior art, which we attacked here, were never *meant* to be similar to one another, but they demonstrably *are* (which is why our attack works). If the field considers it reasonable to study widely available problems/datasets such as ImageNet in this context, then there is a serious conversation to be had around what constitutes a reasonable threat model. Surrogates are of course widely available in this case.

---

> ### Author Response · Authors · 2021-11-17
> **Performance Comparisons and Parameter Values**
>
> > "Experiments on multiple architectures show that the proposed approach is query efficient while achieving similar success rates compared to recent approaches."
>
> We would emphasise here that the success rates are not just similar but saturated, because the problem as posed is (arguably too) easy, with the underlying issue being the fundamental similarity of the input responses of the nets being studied.
> We view our work partly as a sort of ablation study vs. the prior art, which provides strong evidence that all that matters is straightforward extraction of transferable gradient information from the surrogates, with preference for direct transfer wherever possible. As simple as this point is, we don’t believe that this was understood in this manner prior to our work (which is why other results were not competitive).
>
> > "For example, Simba-ODS performance is very close to that of GFCS"
>
> Is this statement based on the median query count table (Table 1)? We agree that, while GFCS still clearly outperforms SimBA-ODS there in a relative sense (the median query count of GFCS is around half that of SimBA-ODS), both medians are low in an absolute sense. However, the median is only one statistic. Fig. 2, which shows the performance CDFs in full, makes clear the extent to which GFCS outperforms SimBA-ODS on untargeted attacks, particularly in the multi-surrogate case. On targeted attacks, as depicted in Fig. 4, the difference is drastic. Note that Fig. 4 uses log scaling. Given the methods’ two different approaches to transfer, we find the targeted result in particular to be as expected.
>
> > "How is the value of epsilon been chosen as it significantly affects the performance of Simba-ODS?"
>
> Regarding the specific case of SimBA-ODS, we have presented two different epsilon tunings: 0.2 and 2.0. The original value of 0.2 is taken from the original paper/code, and is most likely inherited from the original SimBA work. As it is a fairly inappropriate tuning for this problem, we have also chosen the alternative step length of 2.0 based on the search described in Section A.4 and depicted in Fig. 7 of the appendix. We fix the same value of 2.0 for both GFCS and SimBA-ODS, as it has an identical meaning in both contexts, and is a fair like-for-like comparison.
>
> We typically assume that authors have already presented their own papers in the best light possible, with parameters that they themselves optimised for the evaluation, and we always present results using their recommended settings as one of the (possibly multiple) options. However, as noted in Section A.2 of the appendix, we do sometimes take measures to fix competitors’ implementation issues, *improving* their performance in doing so (as with LeBA’s failure to interpolate between input domains of different sizes). Additionally, as ODS-RGF did not standardise its parameters against P-RGF in its own evaluation (in the manner we have done against SimBA-ODS), we have presented P-RGF using both its own settings and those of ODS-RGF, as discussed in the paper. This is itself a very interesting ablation, given its result.
>
> We will respond to the remaining points about further experimental comparisons shortly.

---

> ### Author Response · Authors · 2021-11-23
> **Results for requested experiments (lower norm bound, CIFAR10, hardened net, ablations, SquareAttack); Bayesian Optimisation Approaches**
>
>
> We have added the results of the requested experiments to the submission pdf (in the main paper or in the Appendix) as follows:
> * Lower norm bound (of 5) (Section A.5).
> * Adversarially trained network (Section A.7).
> * CIFAR10 (Section A.6).
>
> Note also that, as also discussed with reviewer KEn1, the loss-gradient-only ablation figures have been added to Table 1.
>
> Originally, we did not include any comparison against surrogate-free methods such as Square Attack since one of our competitors [2] (ODS) ran relevant experiments in their own paper, and claimed dramatic improvement over Square Attack in the L2-bound case (see Table 5 in that reference). Given that Square Attack does not use a surrogate, we did not find this result surprising. Since Square Attack was outright beaten by ODS-RGF in [2], and since we compare ourselves against ODS-RGF under the same experimental setup, we considered the Square Attack comparison to be redundant in our case.
> Having said that, we feel that this is nevertheless a worthwhile comparison, and have performed this experiment.
> We have revised the pdf of the submission to include SquareAttack in the main comparison of Fig.2 (a,b,c) and also in the comparison on targeted attack of Fig.4.
>
> We are happy to cite the Bayesian optimisation approaches from a conceptual perspective, as part of the related work (along with latent-space methods we have already included such as AutoZOOM). However, we don’t see that these represent a competitive comparison. For one thing, both of the above papers present results only for the L_inf norm, and these results are not directly comparable to ours. Beyond this, the very low query counts achieved by [2] on the 0.05 l_inf bound on ImageNet come at the expense of far higher failure rates than the sort we are discussing here (ranging from about 20% to about 35%): The variant of the method that was eventually published (as “Simple and Efficient Hard Label Black-box Adversarial Attacks in Low Query Budget Regimes”) stays within the hard-label/decision-attack threat model, only comparing against other such methods. The performance of [1] is not comparable to ours or any of our chosen competitors: under the standard l_inf bound of 0.05 on ImageNet, their success rate is 60%, with a median query count of 1247. Note that the “state-of-the-art black-box methods” against which they compare are ZOO, AutoZOOM, and GenAttack. Among l_inf-bounded surrogate/prior-leveraging approaches, TREMBA represents an empirically superior modern approach to either of these methods, in both the untargeted and targeted settings.

---

### Author Response · Authors · 2021-11-16
**re: The Requests for Comparisons Against Gradient-Only Methods**

Given that three of the reviewers have raised this in different ways, we have to apologise for our lack of clarity on the following point, which we will amend the text to emphasise:

Part of the point of the demonstration in Fig. 3 is to compare against the gradient-based methods which GFCS subsumes by construction. As can be seen by inspecting the algorithm, the coimage sampling procedure is used *if and only if all surrogate gradients at the current iterate are exhausted*. That is, any input for which GFCS records a nonzero coimage query count represents a failure of the GF portion of the method, a SimBA variation which attempts loss optimisation at fixed-length forward and backward steps in the surrogate gradient directions (within a projection step, in our implementation). This directly provides the “GFCS - ODS” ablation requested by SLVp, and, in different wording, by wR15 and KEn1. Note that this is simply a version of PGD/PGA in which the backward direction is also considered. Note also that FGM is just a special case of PGD/PGA in which the step count is set to 1 and the step size is set equal to the norm bound (FGSM is its $L_\infty$ version, so FGM is the appropriate comparison).

Thus, while we did not provide their counts as a summary statistic in the original submission, all points in Fig. 3 above the zero-coimage-query-count line represent failure cases of gradient transfer without backup (i.e. GFCS - ODS). We give the transfer-only failure rates, for both Fig. 3 and the appendix’s Fig. 6 (where the victims are ResNet-50 and Inception-v3) as follows:
- VGG-16 1-surr: 41.45%
- VGG-16 4-surr: 1.35%
- ResNet-50 1-surr: 24.10%
- ResNet-50 4-surr: 3.5%
- Inception-v3 1-surr: 61.75%
- Inception-v3 4-surr: 19.8%

The marginal histograms in Figs. 3 and 6 communicate this information directly, but issues of scale and size mean that, particularly in the right-hand case, it would have been better had we provided numbers. We originally experimented with log scaling of the marginals’ dependent axis, but it created clarity problems of its own. We will add the failure statistics, as above, to Figs. 3 and 6.

KEn1's note about the right-hand plot in Fig. 3 is well spotted and well observed: it is indeed the case that most (though not all) of the points, given this suite of surrogates against VGG-16, require only the loss gradients, and succeed far quicker by using these than any of the more indirect competitors. In fact, occurrences like this are precisely the sort of thing we ourselves wanted to highlight, in illustrating the importance of not abandoning direct gradient transfer on this problem! However, even in this case, it is evident that a gradient-only approach needlessly leaves failures on the table, albeit a relatively small number (1.35%). GFCS offers the ability to easily succeed on the cloud of points that stray from the gradient-only baseline at the bottom of the figure, which otherwise would have failed. But beyond this, the small failure count in this particular case owes to the fact that VGG-16 (a victim in the P-RGF paper) is an “easy victim” when using the 4-surrogate set from the ODS paper. (Note that this fact makes the relative inefficiency of some of the competitor methods all the more striking.) Please consult Fig. 6 in the appendix to see the results for ResNet-50 and, crucially, Inception-v3. The transfer-only failure rates for those experiments are stated numerically, above. With a larger space budget, we can simply add these figures to the main paper, and make this point explicitly.

As we think that this point is crucial, we would invite early discussion from the reviewers if there is any contention or request for clarification.

---

> ### Comment · Reviewer_KEn1 · 2021-11-19
> **Easy victims make the comparison confusing**
>
> I understand the results in Table 1 and Figure 2/3, and the advantage of GFCS. However, I am still not convinced the current comparison and discussion are sufficient.
>
> Currently, the authors only highlight the median query count in Table 1, which is the main experiment result in the paper, and this will be not enough to show the advantage of GFCS over "GFCS - ODS", i.e., the success rate should be compared on the table. On Inception-V3, the success rate of GFCS is lower than the best success rate by 1.35%/0.6%. Therefore, the differences of the success rate between "GFCS - ODS" and GFCS on some settings (i.e., 1.35% and 3.5%) are not sufficiently large to state that GFCS outperforms "GFCS - ODS". How can you highlight the success rates in Table 1 to show the advantage of GFCS?
>
> I understand it is because there are "easy victims", but these easy victims make the comparison confusing. In other words, it seems that the current experiment setting containing easy victims is not appropriate to clearly show the advantage.

---

> > ### Author Response · Authors · 2021-11-23
> > **Re: Easy victims make the comparison confusing**
> >
> > We have added the "GFCS - ODS" (i.e. gradient-only ablation) failure rates to Table 1, as we believe they provide very useful clarification of the statistics in the scatter plots. Thanks for this suggestion: this will help to make clearer the two sides of our point that direct gradient transfer is very useful but insufficient to achieve excellent success rates on its own. We have also swapped the version of the experiment where Inception-V3 is used as the victim into the main body of the paper, and put the VGG-16 experiment into the appendix, as we feel this will improve clarity on this point.
> >
> > Regarding the easiness of the two experiments in question, where VGG-16 and ResNet-50 are used as the victims and the set {VGG-19, ResNet-34, DenseNet-121, MobileNet-v2} is used as the surrogate, it should again be stressed that the ResNet-50 version of the experiment *is one of the main experiments in the ODS paper, and the overlooked easiness of that experiment is one of the points we would like to make*. Remember, approaches can fail on both sides: they can rely excessively on direct loss gradient transfer and get stuck (as we show in the above ablation statistics), or they can use surrogate priors (like coimages) too indirectly, leading to inefficiency. The ODS methods are in the latter camp, and we believe that that is worthy of comment, including experimental demonstration. The VGG-16 experiment was done because VGG-16 is one of the main victims studied in the P-RGF paper: we showed what would happen both if the original surrogate choice of ResNet-152 were used as well as the aforementioned set from the ODS paper. The latter case turns out to be extremely "easy". Again, we have rearranged the plots in order to adjust the emphasis.
> >
> > Finally, note that the performance of GFCS is always better than that of "GFCS - ODS" by construction. On any example for which GFCS - ODS succeeds (i.e. on which direct loss gradient transfer is successful on its own), the two algorithms are identical. On all examples for which they are not identical, "GFCS - ODS" fails completely, and all GFCS has to do is succeed using some number of queries within the bound, which it nearly always does. It is true that on "very easy" cases like the "VGG-16 vs. four surrogates" experiment, GFCS - ODS will not be much worse than GFCS. On many other experiments, however, it very much will be, as seen. Note that using Inception-V3 as the victim with ResNet-152 as the surrogate is one of the main experiments in both the P-RGF paper and the LeBA paper, and that the *median* outcome for a loss-gradient-only approach in that case is *failure*, meaning that it has failed to find an adversary for *most* of the cases in this setting.

---

### Author Response · Authors · 2021-11-24
**Summary of changes**

We thank the reviewers for their time, suggestions and feedback.

We have added the comparison and experiments requested by the reviewers as follows:
* Comparisons with SquareAttack [Andriushchenko et al.] are now reported in the plots of Fig.2 (a,b,c) for the untargeted case, and Fig.4 for the targeted case.
* The success rates for the loss-gradient-only ablation (GF, no CS) have been added to Table 1.
* Experiments on CIFAR10 are in Appendix A.6.
* Experiments on an adversarially trained network are in Appendix A.7.
* Experiments on a lower norm bound of 5 are in Appendix A.5.
* We have extended the breakdown analysis of GFCS (Fig.3 and Fig. 6) also to the targeted attack experiment in Appendix A.8.
* We have made small edits to the text (as specified in comments to reviewers) and as required when adding requested figures.

---

### Decision · Program_Chairs · 2022-01-20

**Decision:**

Accept (Poster)

**Comment:**

The paper shows that the transfer attack is query efficient and the success rate can be kept high with the zeroth-order score-based attack as a backup.  Experiments show state-of-the-art results.

Pros:
- Simple method based on a simple idea.
- State of the art performance.

Cons:
- Proposal is a straightforward combination of two methods, and therefore technical contribution is marginal.
- The threat model is easy (surrogate can be trained on the same datasets and use the same loss function) and questionable.  Most of the experimental evidence shows that the research for this threat model is almost saturated (and the problem seems almost solved).

This paper got a borderline score with reviewer's concerns above.  I agree with the authors that the simplest method is best among those performing similarly, but the threat setting considered might be not very realistic as the authors admitted.  I see the proposed method a kind of egg of Columbus in a negative sense.  Namely, the authors found a shortcut to win a game that was created and adopted by the community.  Perhaps this paper would give an impact on the small community and would make the community change the game.  But to give an impact to a general audience, the authors should convince that there are some situations where the analyzed thread model is realistic and therefore the proposed method is really useful.  Or, the authors could adjust the thread model to be more realistic.  Serious discussion on the thread model would be a big plus to the marginal technical contributions.

After discussion with SAC, and PC, our conclusion is that this paper effectively tells the community that the benchmark they are using is too simple, which alone is worthwhile publishing because this may move the community forward (even if the community is small).